# Practical Hamiltonian learning with unitary dynamics and Gibbs states

Andi Gu [1,2,3] ✉, Lukasz Cincio[3] & Patrick J. Coles[3,4]

We study the problem of learning the parameters for the Hamiltonian of a quantum many-body system, given limited access to the system. In this work, we build upon recent approaches to Hamiltonian learning via derivative estimation. We propose a protocol that improves the scaling dependence of prior works, particularly with respect to parameters relating to the structure of the Hamiltonian (e.g., its locality $k$). Furthermore, by deriving exact bounds on the performance of our protocol, we are able to provide a precise numerical prescription for theoretically optimal settings of hyperparameters in our learning protocol, such as the maximum evolution time (when learning with unitary dynamics) or minimum temperature (when learning with Gibbs states). Thanks to these improvements, our protocol has practical scaling for large problems: we demonstrate this with a numerical simulation of our protocol on an 80-qubit system.

An increasingly relevant task for the study of many-body quantum systems is to learn the associated Hamiltonian operator efficiently (i.e., without requiring resources that scale exponentially in system size). In condensed matter physics, we can experimentally verify our models of quantum materials by comparing theoretical predictions about their effective interactions with the interactions inferred by Hamiltonian learning[1–4]. This verification is also applicable for quantum device engineering. With the expanding capabilities of quantum computers, it is increasingly important to be able to certify their behavior[5], and while benchmarking protocols can give coarse-grained information about a particular quantum device, knowing its Hamiltonian can be significantly more powerful, allowing us to design improved devices[6–8] or better understand the physical origin of failure modes[9–11].

Several promising approaches have been proposed for the Hamiltonian learning problem. An early work[12] demonstrated that systems with local Hamiltonians can be efficiently characterized without requiring full state tomography, which is costly in terms of accuracy in the trace norm. However, this method was limited in its applicability and found to be prohibitively expensive in general. Subsequent approaches[13–16] successfully employed machine learning on small systems. Nonetheless, these methods lacked rigorous performance guarantees or scaling results that would provide confidence in

their application to larger systems, as their performance on such systems has not been explored beyond limited numerical studies. Additionally, several proposals[17–19] suggested learning the coefficients of the Hamiltonian by solving a system of linear equations, with the coefficient matrix determined by local measurement outcomes. However, the performance of these approaches relies on the spectral gap of the coefficient matrix, which remains poorly characterized. Recent works[20,21] have achieved asymptotically optimal sample complexities, albeit with large constant prefactors that render them impractical in real-world scenarios.

In this work, we propose a protocol for Hamiltonian learning that aims to address these shortcomings. Our protocol is motivated by a major application of Hamiltonian learning, which is the characterization of near-term quantum computers. To accommodate this application, our protocol is designed to make relatively weak assumptions about the nature of the system. Specifically, we assume:

- The Hamiltonian we are interested in learning is sparsely interacting (these are generalizations of $k$-local Hamiltonians; see Definition 2).
- Our interaction with the system is limited to the 'prepare-and-measure' model – that is, we do not require the ability to interact with the system under study via another trusted quantum

[1]Department of Physics, University of California, Berkeley, Berkeley, CA, USA. [2]Harvard Quantum Initiative, Harvard University, Cambridge, MA 02138, USA. [3]Theoretical Division, Los Alamos National Laboratory, Los Alamos, NM 87545, USA. [4]Normal Computing Corporation, New York, NY, USA. ✉ e-mail: andigu@g.harvard.edu

simulator e.g.,[22–24] or make interventions (other than measurement) after initializing the system[25]. Two examples of this prepare-and-measure setup are making measurements on time-evolved states or on Gibbs states (Fig. 1). In these two settings, we assume that we can control the evolution time and the temperature, respectively.

- We can prepare fully separable states and make Pauli measurements.

We note that the practicality of these assumptions depends on the experimental platform. Indeed there are other approaches that impose even more stringent assumptions, such as the restriction of only being able to prepare a single fixed initial state[26], or the ability to make measurements on only a single site[27]. In our work, we do not impose such restricted assumptions, as they do not align with the application we focus on, namely the characterization of near-term quantum computers. In this context, it remains a natural assumption that we have the ability to prepare arbitrary product states and perform Pauli measurements on arbitrary sites. A further advantage of our protocol is that it is easily parallelizable. In short, in this work, we will describe a Hamiltonian learning protocol that requires only $\mathcal{O}(\epsilon^{-2}\text{polylog}(n/\epsilon))$ samples to recover every parameter of a sparsely interacting $n$-qubit Hamiltonian up to an error $\epsilon$. We will conclude by providing a concrete prescription for optimal configurations of the protocol when used in practice, and demonstrate its performance with numerical examples.

## Results

In this work, we will treat the system under study as a black box system with an unknown Hamiltonian $H$, and our goal will be to efficiently infer $H$ with access to only a limited number of inputs to, and outputs from the black box. Importantly, we use the 'prepare-and-measure model' of interaction with our system (see Fig. 1). This model of interaction prohibits any quantum channel between the system under study (whose Hamiltonian we are trying to learn) and some other quantum processing unit. Furthermore, after initializing the system in some state, it prohibits any interaction with the system other than making measurements. Two typical examples of this are Hamiltonian learning using unitary dynamics and Gibbs states. For the former, we initialize the system in some known state $\rho_0$, and evolve it forward in time by $t$, resulting in the state:

$$\rho(t) = e^{-iHt}\rho_0 e^{iHt}. \tag{1}$$

For the latter, we assume we have access to a system in thermal equilibrium at a temperature $\beta^{-1}$. That is, we have access to the Gibbs state

$$\rho(\beta) = \frac{\exp(-\beta H)}{\text{Tr}(\exp(-\beta H))}. \tag{2}$$

We assume that we can control the parameters $t$ and $\beta$, respectively. Finally, we assume that we can measure some observable $P$ of the final states $\rho(t)$ and $\rho(\beta)$. However, we do not insist on arbitrary control over $\rho_0$ and $P$; we only consider the case where $\rho_0$ is fully separable and $P$ is a local Pauli operator.

Using these two interaction models, we propose a method for Hamiltonian learning that relies on a simple intuition. For some particular state preparation and measurement (SPAM) settings (consisting of a prescription for the observable $P$, and in the case of unitary dynamics, the initial state $\rho_0$), which we write as $\mathcal{S}$, we can define a function $f_{\mathcal{S}}$ as the expectation value of $P$ on the state $\rho(t)$ and $\rho(\beta)$:

$$f_{\mathcal{S}}(x) = \begin{cases} \text{Tr}(P\rho(t=x)) & \text{for unitary evolution} \\ \text{Tr}(P\rho(\beta=x)) & \text{for Gibbs states}. \end{cases} \tag{3}$$

We will show that for the appropriate choice of SPAM parameters, $f_{\mathcal{S}}(x)$ can be viewed as black box function in $x$. Using this framework, we describe our basic approach below. For concreteness, we will consider learning with unitary evolution (the analysis for Gibbs states follows similarly in the Supplementary Note 5). First, to assist the reader, we provide below a glossary of notation (Table 1) to serve as reference.

### Preliminaries

To set the stage, we first give a formal definition of the Hamiltonian learning problem and define a sparsely interacting Hamiltonian.

**Definition 1.** (Hamiltonian learning problem). Fix a Hamiltonian on an $n$-qubit system that has an expansion in the Pauli basis:

$$H = \sum_{i=1}^{r} \theta_i P_i, \tag{4}$$

where each $P_i \in \{I, \sigma_x, \sigma_y, \sigma_z\}^{\otimes n}$ is a Pauli operator and $\Theta = [\theta_1, \ldots, \theta_r]^T \in \mathbb{R}^r$ are the Hamiltonian coefficients. We assume the Hamiltonian is traceless (i.e., $P_i \neq I^{\otimes n}$), and that we know the structure of the Hamiltonian (i.e., which Paulis $P_m$ are present in the expansion), but that the coefficients $\theta_m$ are unknown. The Hamiltonian learning problem is to infer all of the coefficients $\theta_m$ up to an additive error $\epsilon \cdot \max_m |\theta_m|$ with success probability at least $1 - \delta$. We will assume two

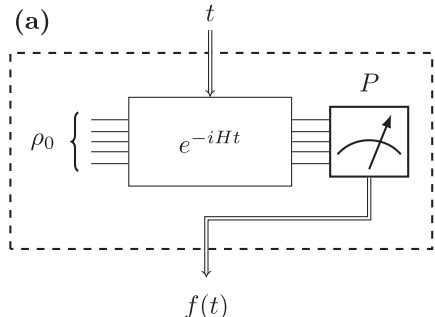
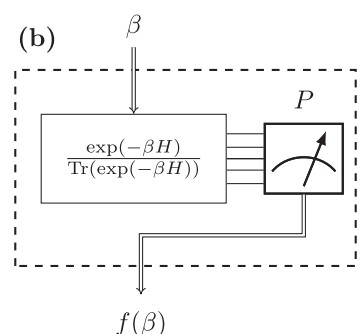

**Fig. 1 | Classical interaction with quantum systems.** The 'prepare and measure' model for interacting with a quantum system. We view the system as a set of oracles indexed by the state preparation and measurement parameters $\rho_0$, $P$ in the time evolution case, and $P$ in the Gibbs state case. These oracles take some input $t$ or $\beta$, and we use their output to characterize the Hamiltonian. In (**a**), we show our model

for time evolution, wherein we control three quantities: $\rho_0$, $t$, and $P$. We assume we can evolve the input state $\rho_0$ forward in time, and after a time $t$, we make a measurement of the observable $P$. In (**b**), we show our model for learning from Gibbs states, wherein we control two quantities $\beta$ and $P$. We assume we have access to the Gibbs state at temperature $\beta^{-1}$, and then measure the observable $P$.

## Table 1 | Glossary of Notations

| Symbol | Definition |
| --- | --- |
| $\epsilon$ | Targeted error (in the $\ell_\infty$ norm) for recovering the Hamiltonian coefficients (Definition 1) |
| $\delta$ | Maximum allowable probability that the recovery (up to an error $\epsilon$) fails (Definition 1) |
| $r$ | Number of coefficients to learn in the Hamiltonian (Definition 1) |
| $\mathscr{D}$ | The degree of the Hamiltonian – measures the connectedness of $H$ (Definition 2) |
| $\tau$ | A typical time scale for the Hamiltonian $H$ (Definition 3) |
| $N$ | Number of samples required in evaluating $f$ at one point (Definition 4) |
| $A$ | Maximum evolution time prescribed by the Hamiltonian learning algorithm (Definition 4) |
| $L$ | Fitting polynomial degree prescribed by the Hamiltonian learning algorithm (Definition 4) |

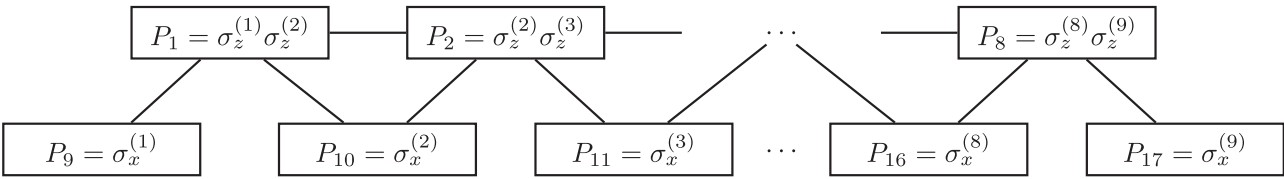

**Fig. 2 | Interaction graph for a transverse field Ising model.** Interaction graph $\mathcal{G}$ for a 9-qubit transverse field Ising model. The degree of this Hamiltonian is $\mathscr{D} = 4$, since for instance $P_2$ is connected to 4 other Pauli terms.

types of data access which define different variants of the Hamiltonian learning problem.

1. **Unitary evolution:** We can prepare the system in some initial product state $\rho_0$ and evolve for a specifiable duration of time $t$. We can then make a measurement of some local Pauli observable on this time-evolved state.
2. **Gibbs states:** We can prepare the system in a Gibbs state at some specifiable temperature. We can then make a measurement of some local Pauli observable on this Gibbs state.

**Definition 2.** (Sparsely interacting Hamiltonian). The interaction graph $\mathcal{G}$ (called the "dual" interaction graph by Haah et al.[21]) of a Hamiltonian consists of a set of vertices $V$ and edges $E$.

$$V = \left\{ P_i | i = 1, \ldots, r \right\}, \tag{5}$$

$$E = \left\{ \left( P_i, P_j \right) | \left( \mathrm{supp}(P_i) \cap \mathrm{supp}\left(P_j\right) \neq \varnothing \right) \wedge (i \neq j) \right\}. \tag{6}$$

Each vertex represents one Pauli operator $P_i$ in the Hamiltonian, and there are edges between two vertices if the support of their corresponding Pauli operators overlap. The support of a Pauli, $\mathrm{supp}(P)$, is the set of sites that $P$ acts nontrivially on. We also define the degree $\mathscr{D}$ of the Hamiltonian to be the maximum degree of any node in the interaction graph:

$$\mathscr{D} = \max_{v \in V} {}^{\circ}(v). \tag{7}$$

A Hamiltonian is sparsely interacting if $\mathscr{D} = \mathcal{O}(1)$ (that is, $\mathscr{D}$ does not depend on system size). Notably, this class of Hamiltonians includes geometrically $k$-local Hamiltonians, as this locality constraint implies that the number of terms overlapping with any Pauli term is a function of $k$ alone.

**Example 2.1.** In Fig. 2, we show a sample interaction graph for a 9-qubit transverse field Ising model (TFIM), whose Hamiltonian is

$$H = \sum_{i=1}^{8} \sigma_z^{(i)} \sigma_z^{(i+1)} + \sum_{i=1}^{9} \sigma_x^{(i)}. \tag{8}$$

The TFIM will serve as a prototypical example for the rest of this work.

## Collecting necessary data

Writing the Taylor expansion of Eq. (3) $f_S(t) = \sum_{m=0}^{\infty} c_m \frac{t^m}{m!}$, our protocol will focus on extracting Hamiltonian parameters using only the first order coefficient of the Taylor expansion $c_1$. To infer this coefficient, we will need to collect data that allows us to estimate $f_S(t)$. The amount and nature of this data will depend on the higher order derivatives $c_m$. More specifically, together with the desired accuracy of the learning protocol $\epsilon$, a bound on the norm $|c_m|$ will determine the required accuracy for our estimate of $f_S(t)$, the number of different points at which we evaluate the function, and the specific times at which we evaluate it. The scaling we find for $|c_m|$ varies depending on whether we are using unitary dynamics or Gibbs states, and also depends on the assumptions we make about the Hamiltonian (i.e., the structure parameter $\mathscr{D}$, and whether the Hamiltonian is commuting). This bound is a crucial determining factor for the rest of our algorithm. In this work, we find

$$|c_m| \sim \begin{cases} \mathcal{O}(\mathscr{D}^m m!) & \text{for sparsely interacting Hamiltonians using unitary dynamics} \\ \mathcal{O}(\mathscr{D}^m) & \text{for commuting Hamiltonians using unitary dynamics} \\ \mathcal{O}(\mathscr{D}^{2m} m!) & \text{for sparsely interacting Hamiltonians with Gibbs states}. \end{cases} \tag{9}$$

Importantly, due to the structure of the Hamiltonian (i.e., it is sparsely interacting), $|c_m|$ does not depend on the size of the system. This enables our protocol to achieve a sample complexity that scales only polylogarithmically in $n$.

Having characterized the higher order derivatives, we return to $c_1$: as mentioned above, this is the only derivative we are interested in. This is because, with the appropriate SPAM configuration, the first order Taylor coefficient $c_1$ will correspond to exactly one Hamiltonian parameter. More precisely, by expanding $H$ in the Pauli basis, we find that there is always at least one pair $(P, \rho_0)$ such that $c_1 = \mathrm{Tr}(i[H,P]\rho_0)$ corresponds exactly to one of the Hamiltonian coefficients $\theta_m$. However, this approach only allows us to extract one Hamiltonian parameter at a time. It turns out that if we are careful, we can learn entire sets of parameters at once by applying simultaneous measurements. These sets of parameters can be chosen with an efficient classical analysis of the Hamiltonian's interaction graph: the key idea is that if two Pauli terms in the Hamiltonian are far enough apart, they have no effect on each other (to first order in time). After these sets are chosen, we can use a single fixed state $\rho_0$, and a set of commuting observables $\{P_i\}$ such that each

$\text{Tr}(i[H, P_i]\rho_0)$ extracts one Hamiltonian parameter, and all the observables $P_i$ can be measured simultaneously. Furthermore, the observables $P_i$ can be chosen to be single qubit Paulis and the initial state $\rho_0$ will be a fully separable state. The reduced state for each site will be either the maximally mixed state $I/2$ or an eigenstate of $X$, $Y$, or $Z$; the full state $\rho_0$ is a tensor product of these single qubit states. These states are easily prepared from $|0\rangle^{\otimes n}$ by applying a constant number of single qubit gates. This simultaneous measurement technique allows us to learn all the Hamiltonian parameters with a sample complexity that is only logarithmic in the number of parameters.

After the SPAM parameters have been determined, we then evaluate $f_S$ to collect a dataset that will subsequently allow us to infer $c_1$. This dataset collection is the only part of our protocol that requires interaction with the system under study. For Hamiltonian learning with unitary dynamics, this involves initializing the system in a product state $\rho_0$, evolving for some time $t_1$, then measuring the set of observables $P_i$. This is repeated $L$ times for different (predetermined) evolution times $t_1, t_2, \ldots t_L \in [0, A]$ up to some maximum time $A$.

## Classical postprocessing

Having constructed our dataset, our Hamiltonian learning protocol can be summarized as follows. For each Hamiltonian parameter $\theta_i$, we fit the corresponding data in our dataset with a degree $L-1$ polynomial in $t$. The first derivative of this fitted polynomial at $t = 0$ serves as an estimate for the parameter $\theta_i$. The following is an informal sketch of our algorithm.

By using a form of polynomial regression known as Chebyshev regression (which simply consists of choosing $t_\ell$ judiciously), we can guarantee that $c_1$ can be estimated with a bias $\mathcal{O}\left(\frac{A^L |c_L|}{L!}\right)$. If $|c_L|$ grows no faster than a factorial, as is the case in Eq. (9), the bias decreases (at least) as a power law in $L$ for suitably chosen $A$. However, our overall error scaling cannot achieve this bound due to the presence of noise when evaluating $f_S$, as increasing $L$ will result in an increase in the variance of our estimator for $c_1$. The modeling error (bias) must be carefully traded against the effects of noise (variance). By appropriately balancing these two, we show that we are able to achieve almost shot noise-limited performance. This is made precise by the following theorem.

**Theorem 1.** (Hamiltonian learning with unitary dynamics). For the appropriate choice of Chebyshev degree $L \sim \mathcal{O}(\log \epsilon^{-1})$ and evolution time $A \sim \mathcal{O}(1)$, the algorithm shown in Box 1 solves the Hamiltonian learning problem with sample complexity

$$\mathcal{O}\left(\frac{\mathscr{D}^4 \log(r/\delta) \, \text{polylog}(\mathscr{D}/\epsilon)}{\epsilon^2}\right), \tag{10}$$

and classical processing time complexity

$$\mathcal{O}\left(\frac{\mathscr{D}^2 r \log(r/\delta) \, \text{polylog}(\mathscr{D}/\epsilon)}{\epsilon^2}\right). \tag{11}$$

**Proof.** See Supplementary Note 4.

Similar to the results of França et al.[28], this can be generalized, via careful selection of initial states and measurements, to learn the Lindbladian (when expanded in the Pauli basis) of open quantum systems undergoing Markovian dynamics. The sample and classical processing time complexity using Gibbs states is only worse by a factor $\mathscr{D}$ and $\mathscr{D}^2$, respectively.

## Numerical simulations

In Theorem 1, we have established the theoretical sample and processing time complexities of our Hamiltonian learning protocol, indicating its effectiveness under certain settings of the Chebyshev degree $L$ and evolution time $A$. However, to provide practical guidance, we now delve into the optimal configurations of our algorithm for real-world applications. This includes prescribing specific values for $L$ and $A$ based on numerical considerations. Additionally, we present compelling numerical results obtained from an 80-qubit transverse field Ising model (TFIM), providing empirical evidence that further supports the utility of our protocol. Our aim will be to learn the following TFIM Hamiltonian:

$$H = \sum_{i=1}^{n-1} J_i \sigma_z^{(i)} \otimes \sigma_z^{(i+1)} + \sum_{i=1}^{n} B_i \sigma_x^{(i)}, \tag{12}$$

where $J_i$, $B_i$-Unif(−1, 1). We choose the TFIM for its broad range of applications[29], including its relevance for quantum computing platforms such as Rydberg atom arrays[30]. The dynamics of this Hamiltonian are simulated with the time evolution block-decimation method[31–35].

Our protocol has two hyperparameters that determine its performance: the maximum evolution time $A$ and the fitting polynomial degree $L$. Setting these parameters is a delicate balance between noise-induced error and modeling errors. If $A$ is too low or $L$ is too high, the variance in the dataset will dominate the error, and on the other hand, if $A$ is too high or $L$ is too low, the modeling error will dominate. It is generally desirable to set these two parameters such that the modeling and noise errors are comparable. However, in some settings, it may be desirable to let the dataset variance grow somewhat larger than the modeling error, since this error can be quantified exactly (see Supplementary Note 3), where $\sigma_\ell^2$ can be obtained by a bootstrap estimate from the dataset. There are no similar methods to quantify the modeling error. One possible method for setting $A$ and $L$ can be to optimize the error bounds (see Fig. 3). Numerically, these optimal values behave

---

## BOX 1

# Algorithm for Hamiltonian learning with unitary dynamics (informal)

1: **procedure** INFERCOEFFICIENTS ($N, L, A$)
2:     Partition the $r$ Hamiltonian coefficients into $\mathscr{D}^2$ subsets $\{\mathbf{V}_i | i = 1, \ldots, \mathscr{D}^2\}$
3:     **for** each subset $\mathbf{V}_i$ **do**
4:         Define the observables $\{P_j\}$ (one for each coefficient $c_j \in \mathbf{V}_i$) and initial state $\rho_0$ such that $\text{Tr}(i[H, P_j]\rho_0/2) = \theta_j$.
5:         Choose $L$ different times $\{t_\ell \in [0, A] | \ell = 1, \ldots, L\}$ at which to evaluate $f_j(t) \equiv \text{Tr}(P_j e^{-iHt} \rho_0 e^{iHt})$.
6:         For each $\ell$, use an $N$-sample mean estimator to estimate $f_j(t_\ell)$
7:         Fit a degree $L-1$ polynomial to the data $(t_\ell, f_j(t_\ell))$.
8:         $c_1 \leftarrow$ first derivative of the fitted polynomial at $t = 0$.
9:         Output $c_1/2$.

---

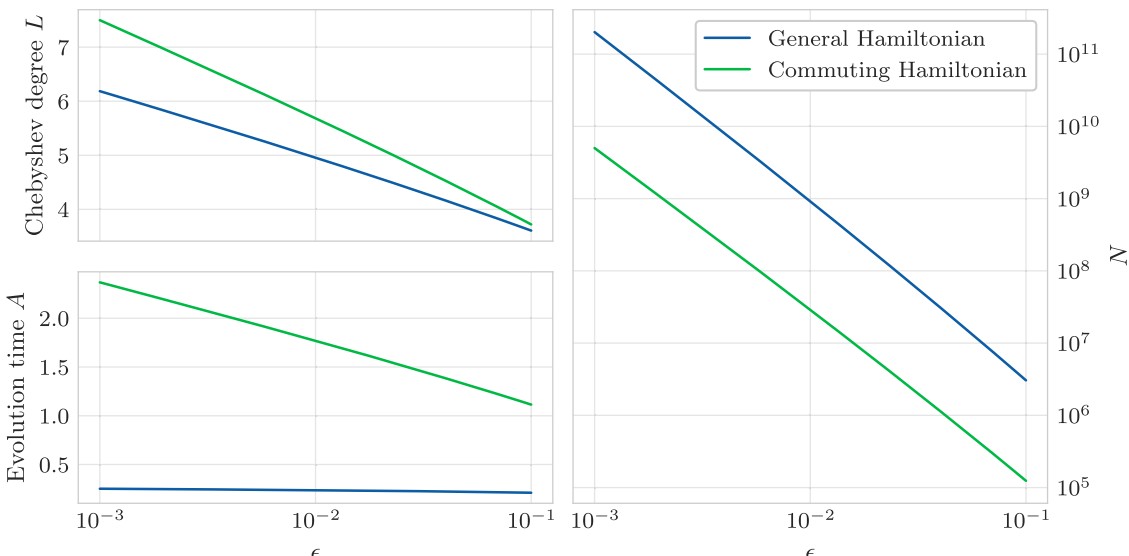

**Fig. 3 | Optimal hyperparameter settings.** Settings for $N$, $L$, and $A$ as a function of the desired error $\epsilon$. These settings are found based on minimizing the upper bound on $N \cdot L$ (in practice, $L$ can only take integer values, so the values shown would be rounded to the nearest integer). For the case of arbitrary Hamiltonians, we observe $L \sim \mathcal{O}(\log \epsilon^{-1})$, $A \sim \mathcal{O}(1)$, and $N \sim \mathcal{O}(\text{polylog}(1/\epsilon)\epsilon^{-2})$. We find similar scaling for the case of the commuting Hamiltonian in every variable except $A$, which also scales as $\mathcal{O}(\log \epsilon^{-1})$. Despite this, the overall sample complexity is only better than the general case by a constant factor.

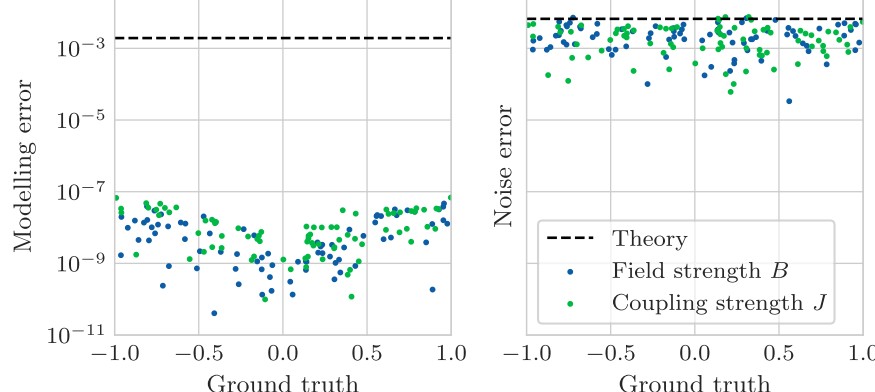

**Fig. 4 | Empirical error of the Hamiltonian learning protocol.** The empirical modeling and noise error of the Hamiltonian learning protocol using the optimal $A$ and $L$ for $\epsilon = 0.01$ as prescribed in Fig. 3. The modeling errors are calculated with a noise-free dataset, and the noise errors are calculated from a single noisy dataset. The dashed line indicates the maximum theoretical modeling error on the left and indicates the predicted variance due to noise on the right.

as anticipated in our theoretical analysis (Theorem 1): the optimal $L^*$ scales with $\mathcal{O}(\log \epsilon^{-1})$, and $A$-1. This leads to a sampling complexity that scales with $\mathcal{O}(\text{polylog}(1/\epsilon)\epsilon^{-2})$.

In Fig. 4, we show the error in the recovered Hamiltonian parameters corresponding to a target error of $\epsilon = 0.01$. As expected, the theoretical prediction for the noise error is close to perfect. However, the modeling error is drastically overestimated by nearly four orders of magnitude. This miscalculated modeling error has important consequences for the algorithm, since it results in a poorly specified evolution time $A$. We propose a number of remedies for this in Supplementary Note 6; the improvements enabled by these techniques are shown in Fig. 5. As demonstrated by the figure, we are able to recover all 159 Hamiltonian parameters up to an error $\lesssim 10\%$ using just ~$10^6$ samples.

## Discussion

In this work, we have discussed the quantum Hamiltonian learning problem. We introduced a unifying model for Hamiltonian learning using both unitary dynamics and Gibbs states. By subsuming these two approaches into the same model, we were able to describe an abstract routine for learning the Hamiltonian of a quantum many-body system given limited access to the system. This routine was based on fixing certain SPAM parameters, then viewing the system as a function $f$ of a single variable. In this work, we consider this variable to be either time $t$ (in which case $f$ represents the time-evolved expectation value of a Pauli observable) or inverse temperature $\beta$ (in which case $f$ represents the thermal expectation value of a Pauli observable). We argued that for the appropriate choice of SPAM parameters, the derivatives of $f$ – particularly $f'(t=0)$ – would correspond exactly to particular coefficients in the Hamiltonian. We then showed that $f'(t=0)$ could be inferred both accurately and efficiently from noisy evaluations of $f$. Finally, we concluded by describing how our protocol could achieve better than linear sample complexity in $r$ (the number of Hamiltonian parameters) by using SPAM configurations amenable to simultaneous measurements.

This culminated in our main result, wherein we proposed an algorithm that achieves an almost noise-limited ($\sim \frac{\text{polylog}(\epsilon^{-1})}{\epsilon^2}$) sample complexity, similar to that of Haah et al.[21] and França et al.[28]. However, our

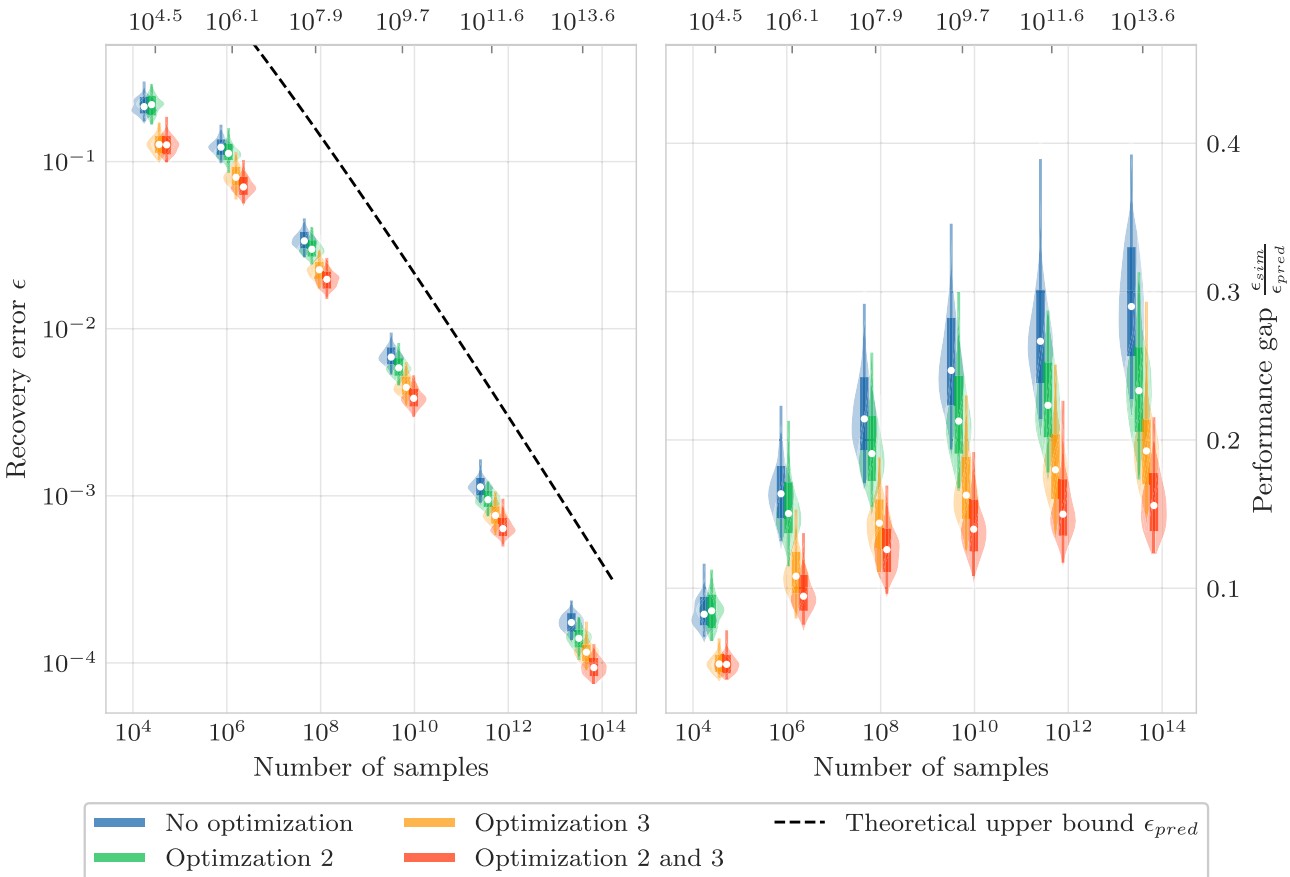

**Fig. 5 | Empirical error distribution.** On the left, we show the maximum absolute error across all 159 coefficients of the 80-qubit TFIM model, plotted against the total number of samples used by the learning protocol, and on the right, we show the quotient of the theoretical error upper bound and the empirical errors from numerical simulations (note the log-log scale for both plots). The violin plots show the distribution of maximum absolute errors from 100 random initializations of the TFIM (with coefficients sampled uniformly between −1 and 1). The distributions show the [1%, 99%] interval in a narrow line, a [16%, 84%] interval in a wider line, and the median marked in white. The violin plots are offset by a small amount for visualization purposes, but each cluster of four violin plots used the same number of queries marked by the dotted gray lines. We set the failure probability to $\delta = 15\%$.

work represents an advance for several reasons. In comparison to Haah et al.[21], we significantly reduce the sample complexity dependence on the parameter $\mathscr{D}$ from $\mathscr{D}^{21}$ to $\mathscr{D}^4$. In comparison to França et al.[28], while their approach includes only Hamiltonian learning from unitary dynamics, our protocol is generalizable to Gibbs states. Furthermore, our approach also offers an additional advantage. Unlike[28], which requires a geometrically local Hamiltonian, our protocol operates efficiently with a "sparsely interacting" Hamiltonian, which is a considerably weaker assumption. This advantage is particularly significant as it eliminates the need for geometric locality. Moreover, we enhance the measurement parallelization overhead from $\mathcal{O}(16^k)$ (assuming a geometrically $k$-local Hamiltonian) to $\mathcal{O}(\mathscr{D}^2)$, a substantial improvement. This is especially relevant in practical applications, where we can often a priori rule out the presence of certain terms in our Hamiltonian from physical constraints or symmetry considerations. That is, oftentimes, we have $\mathscr{D} \ll 4^k$; in these settings, our protocol can provide a significant advantage. Furthermore, by deriving explicit bounds on the performance of our algorithm, we were able to provide precise numerical prescriptions for theoretically optimal hyperparameters such as maximum evolution time and Chebyshev degree. We concluded by proposing a number of heuristic improvements to our algorithm, and argued they were reasonable to apply in general. This combination of improvements makes significant steps towards achieving a practically useful protocol that can be applied experimentally, as indicated by the demonstration of our protocol on a large (80-qubit) simulated problem.

Although we have demonstrated a successful application of our learning algorithm on a simulated problem, this simulation did not include possible detrimental experimental effects. With respect to SPAM errors, our algorithm makes minimal SPAM requirements (requiring only single qubit measurements and simple product states). To the first order, the effect of SPAM errors will only be in the measurement of the first order commutator $\mathrm{Tr}(i[H,P]\rho_0)$. For instance, if our initial state is subject to decoherence, this will result in a systematic underestimate of the Hamiltonian parameters. Therefore, a natural future direction for investigation is how this protocol can be made robust to SPAM errors. Another consideration is the potential discrepancy between the Hamiltonian ansatz used by the learning algorithm and the actual underlying Hamiltonian governing the physical system. In realistic scenarios, the system Hamiltonian may deviate from the assumed form due to various factors such as unaccounted interactions, noise, or experimental limitations. To the first order, terms that are unaccounted for do not affect the performance guarantees of our algorithm except for their effect on $\mathscr{D}$. However, as noted previously, a good estimate of $\mathscr{D}$ is a strong determining factor in the practical performance of our protocol; further investigation is needed to understand the extent to which model mismatches adversely affect performance in practice. We also leave for later works a study of how this protocol can be improved by making stronger assumptions on either the Hamiltonian or the suite of interactions available to us. For instance, we already showed a constant (but significant) drop in the number of measurements required for learning a commuting Hamiltonian with unitary

---

## BOX 2

# Algorithm for estimating the first derivative $\mathrm{Tr}((i[H,P])\rho_0)$

1: **procedure** ESTIMATEDERIVATIVE($N, L, A; P, \rho_0$)
2: **for** $\ell \leftarrow 1,\ldots, L$ **do** ▷ Construct the dataset $\mathcal{D}$ (Definition 4)
3: $t_\ell \leftarrow \frac{A}{2}\left(1 - \cos\left(\frac{2\ell-1}{L}\pi\right)\right)$
4: $y_\ell \leftarrow$ estimate of $\mathrm{Tr}(Pe^{-iHt_\ell}\rho_0 e^{iHt_\ell})$ ▷ Average $N$ measurement outcomes of $P$
5: Fit the coefficients $\tilde{c}_k$ in $\sum_{k=0}^{L-1} \tilde{c}_k \frac{t^k}{k!}$ to the data $\{(t_\ell, y_\ell)|\ell = 1,\ldots,L\}$
6: **return** $\tilde{c}_1$

---

dynamics. We expect a similar effect for Hamiltonian learning with Gibbs states. Furthermore, if we assume we can interact with our system using a trusted quantum simulator of our own, a variety of approaches become possible. Among these is Hamiltonian learning with Loschmidt echoes, as done in Wiebe et al.[22]. Rigorous performance bounds have not yet been found for this approach, but we speculate that a similar application of our techniques may yield improved performance – however, we leave this for future works.

## Methods

In this section, we will describe our derivative estimation protocol, and show that this allows us to make guarantees on the error. First, we establish an elementary procedure for estimating the first order derivative $f'(0)$ given access only to noisy estimates of $f$. We then apply this procedure to Hamiltonian learning with unitary dynamics and Gibbs states.

### Inferring the first-order commutator

For a system evolving under a Hamiltonian $H$ and an initial state given by some density matrix $\rho_0$, the expectation value of any operator $P$ can be written as:

$$\langle P(t)\rangle = \mathrm{Tr}(P\rho_0(t)) = \mathrm{Tr}(Pe^{-iHt}\rho_0 e^{iHt}) = \sum_{m=0}^{\infty} \frac{(it)^m}{m!}\mathrm{Tr}([H^m P]\rho_0), \quad (13)$$

$$\text{where } [H^m P] = \underbrace{[H,[H,\ldots,[H}_{m \text{ times}},P]\ldots]] \text{ with } [H^0 P] = P. \quad (14)$$

This equality is simply using the Heisenberg expansion of the time-evolved operator $P(t)$.

In this section, we define a critical subroutine of our Hamiltonian learning algorithm that infers the expectation $\mathrm{Tr}((i[H,P])\rho_0)$, for $P$ being a local Pauli operator, by measuring time-evolved expectation values. The main idea behind our algorithm is that $\mathrm{Tr}((i[H,P])\rho_0)$ is the time derivative of the expectation $\mathrm{Tr}(Pe^{-iHt}\rho_0 e^{iHt})$. More specifically, the Heisenberg expansion in Eq. (13) expresses the time-evolved expectation of an observable as

$$\langle P(t)\rangle = \sum_{m=0}^{\infty} \frac{i^m}{m!}\mathrm{Tr}([H^m P]\rho_0)t^m. \quad (15)$$

Therefore $\langle P(t)\rangle$ can be modeled as a univariate power series in time, $\sum_{m=0}^{\infty} c_m t^m$, with coefficients

$$c_m = \frac{i^m}{m!}\mathrm{Tr}([H^m P]\rho_0). \quad (16)$$

If we were able to access $\langle P(t)\rangle$ exactly, the most effective way to find $c_1$ would be to simply differentiate $\langle P(t)\rangle$ via finite differences with very small $\Delta t$ (i.e., $c_1 \approx \frac{\langle P(\Delta t)\rangle - \langle P(0)\rangle}{\Delta t}$). Since our measurements of $\langle P(\Delta t)\rangle$ are subject to shot noise, the variance of this estimator scales with $\mathcal{O}((\Delta t)^{-2})$, preventing us from using arbitrarily small $\Delta t$. However, as $\Delta t$ grows, the bias in the finite difference estimator grows. The algorithm in

Box 2 is a generalization of finite differencing, and uses Chebyshev regression (see Supplementary Note 1) to estimate $c_1$. This algorithm takes as input a maximum evolution time $A$ and an cutoff degree for the Chebyshev polynomial $L$. This finite cutoff degree induces biases in the recovered polynomial coefficients, however, we will demonstrate that this bias is suppressed much more effectively than for the finite-difference estimator, as it turns out that these errors scale in a power-law with power $L$. As mentioned in the beginning of this section, this error bound depends on a bound for the derivative $|\frac{d^l \langle P(t)\rangle}{dt^l}| = |\mathrm{Tr}([H^m P]\rho(t))|$. Since $\rho(t)$ is a density matrix, a simple application of the Höelder inequality shows that $|\mathrm{Tr}([H^m P]\rho(t))| \leq |[H^m P]|$ (where $|\cdot|$ denotes the spectral norm). We can bound spectral norms of iterated commutators with the Hamiltonian as follows:

**Definition 3.** (Typical scales). We define a typical time scale

$$\tau = \frac{1}{2\mathscr{D}|\Theta|_\infty} \quad (17)$$

of our Hamiltonian. The appearance of $|\Theta|_\infty$ in these scales is unsurprising; scaling all the coefficients up by some constant factor will decrease the time scale of the time evolution by the same factor. The structure parameter $\mathscr{D}$ appears in this time scale because, all things being equal, we expect a highly connected Hamiltonian to have observables that change faster compared to weakly connected ones. Indeed, in Supplementary Note 2, we show an upper bound for the norm of the $m$th iterated commutator between $H$ and $P$ scales roughly with $\sim\tau^{-m}$.

**Definition 4.** (Dataset). Assume we are given the following hyperparameters:

- $L$, which tells us how many different times at which to evaluate $\langle P(t)\rangle$;
- $A$, which tells us the maximum time at which we want to evaluate $\langle P(t)\rangle$; and
- $N$, which tells us how many samples we use to estimate a single evaluation of $\langle P(t)\rangle$.

We construct the dataset $\mathcal{D}$ by evaluating $\langle P(t)\rangle$ at the roots $\{z_i|i=1,\ldots,L\}$ of the $L$th Chebyshev polynomial (see Supplementary Note 1 for a review of Chebyshev polynomials). Our dataset comprises of $L$ points:

$$\mathcal{D} = \{(t_1,y_1),(t_2,y_2),\ldots,(t_L,y_L)\}, \text{ where}$$
$$t_i = \frac{A}{2}(1+z_i), \quad (18)$$
$$y_i \sim Y_i,$$

where $Y_i$ is an $N$-sample mean estimator of $\langle P(t_i)\rangle$. That is, it satisfies $\mathbb{E}[Y_i] = \langle P(t_i)\rangle$ and $\mathrm{var}[Y_i] = \sigma_i^2/N$, where $\sigma_i^2$ is the variance for a single measurement of $\langle P(t_i)\rangle$. The mapping $t_i = \frac{A}{2}(1+z_i)$ ensures that the evolution time is nonnegative and never exceeds $A$.

Having collecting the dataset, it is simple to infer the first derivative $c_1$.

The following theorem shows that for the appropriate choice of evolution time $A$ and Chebyshev degree $L$, the error of the estimator $\tilde{c}_1$ in Box 2 is close to being noise-limited.

**Theorem 2.** (Sample complexity for one coefficient). Fix some maximum failure probability $\delta$ and an error $\epsilon$. Assume that we have access to an unbiased (single-shot) estimator of $\langle P(t) \rangle$ with variance $\sigma^2 \leq 1$. Furthermore, assume $|P| \leq 1$. Then there is some choice of maximum evolution $A \sim \tau$ and Chebyshev degree $L \sim \log \epsilon^{-1}$ such that with

$$N = \mathcal{O}(\log(1/\delta)\text{polylog}(1/\epsilon)\epsilon^{-2}) \qquad (19)$$

sample complexity, we can construct an estimator $\tilde{c}_1$ such that $|c_1 - \tilde{c}_1| \leq \epsilon \cdot \mathscr{D}$, except with a failure probability at most $\delta$.

**Proof.** See Supplementary Note 3. □

### Recovering Hamiltonian coefficients

With an efficient algorithm for accurately estimating first order commutators $\text{Tr}(i[H,P]\rho_0)$, it is possible to construct an algorithm that can infer the coefficients of $H$ using these commutators. The idea is to carefully choose $\rho_0$ and $P$ so that $\text{Tr}(i[H,P]\rho_0)$ corresponds to one parameter at a time.

First, we introduce the notation that $\rho_0^{(\mathcal{X})}$ and $P^{(\mathcal{X})}$ will be the reduced state or Pauli matrix (respectively) that is restricted to the qubits in $\mathcal{X}$, and $\mathcal{X}'$ will be the set of all qubits not in $\mathcal{X}$.

**Lemma 1.** (Term selection) Let $P$ be some Pauli operator such that there exists some $i \in \{1, \ldots, r\}$ where $\text{supp}P \subseteq \text{supp}P_i$ and $\frac{i[P_i,P]}{2} \neq 0$. Let

$$\mathcal{X} = \text{supp}P_i, \qquad (20)$$

$$\mathcal{Y} = \left( \bigcup \{ \text{supp}P_j | \text{supp}P_j \cap \mathcal{X} \neq \varnothing \} \right) \setminus \mathcal{X}, \qquad (21)$$

$$\mathcal{Z} = (\mathcal{X} \cup \mathcal{Y})', \qquad (22)$$

$$\rho_0 = \left( \frac{\mathbb{I} + i[P_i,P]/2}{2^{|\mathcal{X}|}} \right)^{(\mathcal{X})} \otimes \left( \frac{\mathbb{I}}{2^{|\mathcal{Y}|}} \right)^{(\mathcal{Y})} \otimes \rho_0^{(\mathcal{Z})}. \qquad (23)$$

In words, $\mathcal{Y}$ is a neighborhood around $\mathcal{X}$ that contains the support of all Paulis that intersect with $\mathcal{X}$, and $\mathcal{Z}$ is the set of all qubits that are not in $\mathcal{X} \cup \mathcal{Y}$. The state $\rho_0$ is defined such that for all qubits in $\mathcal{Y}$, it is the maximally mixed state and for qubits inside $\mathcal{X}$, $\rho_0$ is defined in a way such that $\text{Tr}(i[P_i,P]\rho_0^{(\mathcal{X})}/2) = 1$, and for all other qubits, $\rho_0$ can be anything. Then:

$$\text{Tr}(i[H,P]\rho_0) = \theta_i. \qquad (24)$$

**Proof.** See Supplementary Note 4. □

This defines a simple algorithm for Hamiltonian learning. For simplicity, for any Pauli $P_i$, we will simply set the observable $P$ to be a single qubit Pauli acting on one site in $\mathcal{X}$ such that $[P_i, P] \neq 0$ (see Box 3).

However, the runtime of this algorithm is $\Omega(r)$, since this procedure must be called once for each term in the Hamiltonian. We propose an improvement of this algorithm wherein we estimate $\text{Tr}(Pe^{-iHt}\rho_0 e^{iHt})$ for many different choices of $P$ simultaneously. We aim to set $\rho_0$ in such a way that we can extract coefficients for many terms simultaneously. Yet, rather than using shadow tomography (as done in França et al.[28]), which can result in $\mathcal{O}(16^k)$ scaling, we carefully take advantage of our knowledge about the Hamiltonian structure to get a smaller parallelization overhead. The way forward relies on the fact that in Lemma 1, $\rho_0^{(\mathcal{Z})}$ can be anything. Similarly to Haah et al.[21], we partition the terms of our Hamiltonian into groups of terms that can each be inferred simultaneously. This partition is based on a graph coloring; for details, see the Supplementary Note 4.

**Definition 5.** (Squared graph). Let the square of the interaction graph, $\mathcal{G}^2$, be the graph with the same vertex set as $\mathcal{G}$ and in which any two vertices are connected if their distance in $\mathcal{G}$ is at most 2. In words, the edges for $\mathcal{G}^2$ are

$$\left\{ (i,k) | \exists j \left( \text{supp}P_i \cap \text{supp}P_j \neq \varnothing \right) \wedge \left( \text{supp}P_j \cap \text{supp}P_k \neq \varnothing \right) \wedge (i \neq k) \right\} \qquad (25)$$

Our algorithm will rely on a graph coloring of $\mathcal{G}^2$. The essential idea is that for Paulis of the same color, there is always a "moat" separating them. This moat will then be filled with maximally mixed states, which completely suppresses the influence of terms that we are not interested in. A partitioning of the Hamiltonian terms via some $C$-coloring of $\mathcal{G}^2$ makes it natural to rewrite the Hamiltonian using a double sum notation:

$$H = \sum_{i=1}^{C} \sum_{j=1}^{|\mathbf{v}_i|} \theta_{i,j} P_{i,j}, \qquad (26)$$

where $\mathbf{v}_i$ is the set of all Paulis with the same color $C_i$. For instance, see Supplementary Fig. 3 for a coloring of the squared interaction graph for a 9-qubit TFIM.

**Lemma 2.** (Simultaneous inference for a partition) Let $\mathbf{v}_i$ be a partition in a coloring of $\mathcal{G}^2$. The coefficient for each Pauli in $\mathbf{v}_i$ can be inferred with up to an error $\epsilon |\Theta|_\infty$, with failure probability for each individual coefficient being at most $\delta$ (so the overall failure probability is upper bounded by $\delta \cdot |\mathbf{v}_i|$). This can be done with sample complexity

$$\mathcal{O}(\mathscr{D}^2 \log(1/\delta)\text{polylog}(\mathscr{D}/\epsilon)\epsilon^{-2}). \qquad (27)$$

**Proof.** See Supplementary Note 4. □

---

## BOX 3

# Algorithm for Naive Hamiltonian learning

1: **procedure** NAIVEINFERCOEFFICIENTS $(\tau, L, A, N)$
2:    **for** $i \leftarrow 1, \ldots, r$ **do**
3:       $P \leftarrow$ single qubit Pauli acting on one site in $\mathcal{X}$ where $[P_i, P] \neq 0$
4:       $\rho_0 = \left( \frac{\mathbb{I} + i[P_i,P]/2}{2^{|\mathcal{X}|}} \right)^{(\mathcal{X})} \otimes \left( \frac{\mathbb{I}}{2^{|\mathcal{Y}|}} \right)^{(\mathcal{Y})} \otimes \rho_0^{(\mathcal{Z})}$                     ▷ $\rho_0^{(\mathcal{Z})}$ is any density matrix
5:       $\tilde{\theta}_i \leftarrow$ ESTIMATEDERIVATIVE $(N, L, A; P, \rho_0)$                     ▷ See Box 2
6:    **return** $\tilde{c}_1$

---

---

**BOX 4**

# Algorithm for Hamiltonian learning with unitary dynamics

---

1: **procedure** PARTITIONINFERCOEFFICIENTS($\tau,\mathcal{G},L,A,N$)
2: $\{\mathbf{V}_i\} \leftarrow$ GRAPHCOLOR($\mathcal{G}^2$) ▷ Find $\mathcal{D}^2 + 1$ partitions of $\mathcal{G}^2$
3: **for** $i \leftarrow 1,\dots,\mathcal{D}^2 + 1$ **do**
4: **for** $j \leftarrow 1,\dots,|\mathbf{V}_i|$ **do** ▷ Define the observables and states (Lemma 2)
5: $P'_j \leftarrow$ a single-qubit Pauli such that $\left[P'_j P_j\right] \neq 0$
6: $\rho_0^{(\mathrm{supp}P_j)} = (\mathbb{I} + i\left[P_j P'_j\right]/2)/2$
7: **for** $q \in (\mathrm{supp}\mathbf{V}_i)^c$ **do** ▷ Fill the moats
8: $\rho_0^{(q)} \leftarrow \mathbb{I}/2$
9: **for** $\ell \leftarrow 1,\dots L$ **do** ▷ Construct the dataset (Definition 4)
10: $t_\ell \leftarrow \frac{A}{2}(1 - \cos(\frac{2i-1}{L}\pi))$
11: $M_\ell \leftarrow N$ measurements of $\mathrm{Tr}(P'_j e^{-iHt_\ell}\rho_0 e^{iHt_\ell})$ for $j \in \{1,\dots,|\mathbf{V}_i|\}$
12: **for** $j \leftarrow 1,\dots,|\mathbf{V}_i|$ **do** ▷ Estimate the first commutator for each Pauli in $\mathbf{V}_i$
13: $y_\ell \leftarrow$ estimate of $\mathrm{Tr}(P'_j e^{-iHt_\ell}\rho_0 e^{iHt_\ell})$ by averaging over $M_\ell$
14: Fit the coefficients $\tilde{c}_k$ in $\sum_{k=0}^{L-1} \tilde{c}_k \frac{t^k}{k!}$ to the data $\{(t_\ell,y_\ell)|\ell=1,\dots,L\}$
15: $\tilde{\theta}_{i,j} \leftarrow \tilde{c}_1$ ▷ Estimate for the coefficient $\theta_{i,j}$

---

**Theorem 3.** (Hamiltonian learning with unitary dynamics). Fix a sparsely interacting Hamiltonian $H$ that has $r$ terms in its Pauli expansion with coefficients $\Theta$. For the appropriate choice of Chebyshev degree $L$ and evolution time $A$, the algorithm in Box 3 and Box 4 solves the quantum Hamiltonian learning problem (with an additive error $\epsilon|\Theta|_\infty$ and failure probability at most $\delta$) with sample complexity

$$\mathcal{O}\left(\frac{\mathcal{D}^4 \log(r/\delta)\mathrm{polylog}(\mathcal{D}/\epsilon)}{\epsilon^2}\right), \tag{28}$$

and classical processing time complexity

$$\mathcal{O}\left(\frac{\mathcal{D}^2 r \log(r/\delta)\mathrm{polylog}(\mathcal{D}/\epsilon)}{\epsilon^2}\right). \tag{29}$$

**Proof.** We partition our Hamiltonian terms into sets that can be simultaneously inferred. There are at most $\mathcal{D}^2$ of these sets (for a proof, see Supplementary Note 4) – moreover, this partitioning into at most $\mathcal{D}^2$ can be found with classical greedy algorithm that has runtime $\mathcal{O}(\mathcal{D}^2)$[36]. Now, we apply Lemma 2 to each of these sets. For the detailed proof, see Supplementary Note 4. □

 In a different setup, we may be given access to copies of a Gibbs state at a temperature $\beta^{-1}$. If we measure an observable $P_i$, the expectation will be

$$\langle P_i \rangle_\beta = \frac{\mathrm{Tr}(P_i \exp(-\beta H))}{\mathrm{Tr}(\exp(-\beta H))} \tag{30}$$

In what follows, we apply the analysis of Haah et al.[21] to formulate $\langle P_i \rangle_\beta$ as a polynomial in $\beta$, in accordance to the framework in Eq. (3). We will show that we can learn the coefficients of the Hamiltonian from the first order term in this polynomial, therefore mapping the problem of Hamiltonian learning from Gibbs states onto Hamiltonian learning with unitary dynamics.

**Theorem 4.** (Hamiltonian learning with Gibbs states). The Hamiltonian learning problem (with an additive error $\epsilon|\Theta|_\infty$ and failure probability at most $\delta$) can be solved using

$$\mathcal{O}\left(\frac{\mathcal{D}^5 \log(r/\delta)\mathrm{polylog}(\mathcal{D}/\epsilon)}{\epsilon^2}\right) \tag{31}$$

copies of the Gibbs state. This can be achieved with a time complexity

$$\mathcal{O}\left(\frac{\mathcal{D}^4 r \log(1/\delta)\mathrm{polylog}(\mathcal{D}/\epsilon)}{\epsilon^2}\right). \tag{32}$$

**Proof.** The protocol is a near mirror image of the Hamiltonian learning protocol using unitary dynamics. For the full proof, see Supplementary Note 5. □

## Data availability
The simulation data used to produce Fig. 5 has been deposited in Zenodo[37].

## Code availability
The code for the 80-qubit TFIM numerical simulation can be found on Github[37].

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

## Acknowledgements

The authors thank Hsin-Yuan Huang (Robert), Matthias C. Caro, Diego García-Martín, Marco Cerezo, Susanne Yelin, and Hong-Ye Hu for inspiring discussions and for comments on earlier drafts. A.G. acknowledges support from the U.S. Department of Energy (DOE) through a quantum computing program sponsored by the Los Alamos National Laboratory (LANL) Information Science & Technology Institute. L.C. was supported by the Laboratory Directed Research and Development (LDRD) program of LANL under project number 20210116DR. P.J.C. was supported by the LANL ASC Beyond Moore's Law project.

## Author contributions

A.G. conceptualized, conducted the formal analysis, and wrote the original draft. L.C. and P.J.C. helped in the initial writing, as well as revising and editing the manuscript.

## Competing interests

The authors declare no competing interests.
