## [Peer Review File · Nature Communications]

Practical Hamiltonian Learning with Unitary Dynamics and Gibbs StatesREVIEWER COMMENTS

Reviewer #1 (Remarks to the Author):

Summary:

The authors study the problem of learning a sparsely interacting Hamiltonian from measured data. The Hamiltonian is assumed to have a linear expansion in terms of (known) multi-qubit Pauli operators. The task is to learn the unknown coefficients in this expansion. The experimental scheme consists of evolving a product state under the Hamiltonian in question for different times and performing Pauli basis measurements. Another variant of the scheme assumes Pauli basis measurements on the associated Gibbs state at different inverse temperatures.

The authors' approach applies Chebyshev regression of a 'small degree' to the time series data of certain Pauli observables. The choice of observables allows one to infer the Hamiltonian coefficients as the first-order Taylor coefficients of the time series. By carefully analysis of the simultaneous measurability of different Pauli observables for the coefficients of a sparsely interacting Hamiltonian, the authors reduce the number of measurement settings and, thus, sample complexity. Combining the analytical guarantees for Chebyshev regression with the exponential confidence of a sub-Gaussian median-of-means estimator, the authors show an overall sample complexity of $D^4/\epsilon^2 \text{polylog}(D/\epsilon) \log(r)$ for estimating all r coefficients of a Hamiltonian with degree- D interaction graph to additive precision ϵ . The time complexity of the classical post-processing is also found to be efficient.

In addition, the authors demonstrate the feasibility of their algorithm in numerical simulations of a 80 qubit transverse-field Ising Hamiltonian.

Assessment:

I find the work a valuable contribution to the field. Hamiltonian learning is indeed a fundamental and technologically increasingly relevant task. The authors' approach neatly reduces the amount of samples required for Hamiltonian learning by exploiting the simultaneous measurability of local observables. Using Chebyshev regression instead of finite differences improves robustness and gives analytically well-controlled errors in terms

of the polynomial degree and the maximal evolution time. The manuscript is well-written and clear in its presentation. The technical results and proofs in the supplemental material seem sound and correct.

One might argue that the authors' scheme mostly combines multiple known techniques and that the resulting performance guarantees have little surprise value. As such they constitute mostly technical advances that are predominantly of interest for people working on rigorous guarantees for quantum system identification. For this reason, I am not entirely sure if the Nature Communications are the right venue for this work. Having said this, I personally consider the achieved improvement over existing guarantees an important contribution and in principle support a publication in the Nature Communications after a revision.

My main point of criticism is that the authors are imprecise and sloppy when contextualizing their contribution (in particular, in the introduction and title). In my opinion, the authors unnecessarily stretch the notion of “practicality” and of a “black-box”. I will explain my criticism below together with further remarks. I would like to encourage the authors to address these points in a revision and present their contribution more carefully.

“Practicality”:

I think one should make a clear distinction between demonstrating that a protocol is practical or deriving performance guarantees for a protocol with 'practical scaling'. Furthermore, if one does not carefully compare a lower bound for one protocol with an upper bound of another protocol, it is up to debate if one achieved a practically relevant improvement or demonstrated differences in the available proof techniques for the protocols.

When stating their contribution the authors constantly conflate these two notions to argue for the practicality of their scheme. I agree that the authors' scheme seems to be an experimental practical proposal and the scheme comes with 'practical scalings'. However, while the latter point is clearly demonstrated in the manuscript, the former is much less

obvious to me.

To give concrete examples where I consider the manuscript not sufficiently precise when comparing to existing results:

* What does it mean to have "scaling results that would allow them to be applied on larger systems"? What exactly limits the application of the techniques of Ref. [13-16] to larger systems? Is their computational complexity problematic or were they simply not demonstrated to succeed?

* The authors criticize Refs. [20, 21] for having "constant prefactors that renders them infeasible in practice". Are these prefactors only in the theoretical guarantees and potentially a proof artifact or do they actually determine the practical performance? Is this clearly demonstrated somewhere? When stating their own results in big-O notation the authors also suppress all constant prefactors. So in what sense are they improving? One might even argue that for intermediate system sizes also median-of-mean adds an impractical prefactor that increases the sample complexity by two orders of magnitude.

* When comparing to Ref. [27] in the conclusion it is not clear what k is.

Regarding the claim of experimental practicality: In my experience, practicality very much depends on the platform, especially when probing an only partly controlled system or an analog simulator. It is a priori not clear if interleaving the time evolution with quenches performing Pauli rotations is prohibitively impractical. At the same time, precisely controlling the transition from state preparation to Hamiltonian dynamics and to a measurement in the end has already been found challenging in experiments. In analogy, in the context QPT, SPAM errors have turned out to severely limit the practicality of many schemes.

The numerical demonstration on a 80-qubit transverse field Ising model is a convincing proof-of-concept for the scalability of the authors' method. But the numerical study is too narrow, does not compare against other methods or systematically studies the effect of

experimental imperfections to justify the claim of practicality of the scheme.

For these reasons, I think the actual practicality in an experiment of the authors' protocols is not convincingly demonstrated (as also the authors discuss at the end of the conclusion to some extent). At the same time, the authors' work can nonetheless rightfully claim practical relevance.

“Black-box” and “classical interaction with the system”:

I find calling the setting a “black-box model” “restricted to classical interaction” misleading. The authors assume accurate control of quantum input (product states) and specific local measurements at the end. In the context of QPT this is commonly referred to as a “prepare-and-measure” setup (say, without entangled input or entangling measurements). The Hamiltonian learning setting additionally assumes control over the evolution time (or inverse temperature). While certainly realistic on many quantum devices today, this setup is still far away from a black-box model with only classical interaction to my understanding. Especially, in the context of Hamiltonian learning it is quite natural to assume settings with considerably more limited control over the input states and the accessible measurements. In fact, this has already motivated an entire series of work by Burgarth and co-authors in the past on inferring Hamiltonians under limited access to the system, for example only measurements on one single local site. The work by Wilde et al. 2022 only requires local measurements but only one initial state, to mention another example.

Further remarks and questions:

* To my understanding many 'natural' Hamiltonians might have multiple scales. For example, coupling terms might be orders of magnitude suppressed compared to the free dynamics. How does the scheme perform in such scenarios?

* page 2, first line: I think there is a crucial typo in the stated scaling and it should be $1/\epsilon^2$.

* page 1, Even though a common place, saying that QST is exponentially expensive is a bit misleading in the context of Hamiltonian learning. The authors' guarantee provides a precise estimate in ℓ_∞ -norm (vector max-norm) for the Hamiltonian parameters. Using the same norm, the entries of a quantum state can also be estimated from $\text{polylog}(d)$ state copies using a maximal set of MUBs [Morris & Dakic 2019]. The same holds for spectral norm estimates [Guta et al. 2020]. Only the translation to trace-norm introduces an exponential scaling in the system size.

* page 3: When the structure parameter D is first mentioned and used for stating results in Eq. (4) it is not clear what it actually refers to.

* page 3 typo "initializing the system <in> a product state"

* As far as I know, the first author's surname of Ref. [27] is Stilck-Franca

* page 10, missing reference in proof of Thm. 3

Reviewer #2 (Remarks to the Author):

For any Hamiltonian that is a linear combination of Pauli strings, this work provides an efficient protocol for the estimation of its coefficients via measuring expectation values of time-evolved observables. More specifically, the initial states are certain maximally mixed states in some stabilizer subspaces specifically designed for the task. For the time evolution, the Hamiltonian is applied for certain short time steps. After that, it is sufficient to measure single-qubit Pauli observables. It is proven that the sampling complexity of the estimation protocol scales as (31) with classical postprocessing time (32) where

- D is a certain degree of the Hamiltonians interaction graph in the Pauli basis,
- $1-\delta$ is the confidence lower bound for the estimation

- ϵ the maximum ℓ_∞ -norm error of the Hamiltonian coefficients, and
- I cannot find r again while now writing the report.

In particular, these costs are independent of the number of qubits.

Moreover, it is shown that the scaling can be improved for commuting Hamiltonians and extended to imaginary time, i.e., to the estimation of thermal states with control in the inverse temperature instead of time. The protocol is also practically applicable, as shown with numerical simulations.

Technically, the protocol relies on approximating the expectation values of the time-evolved observables $\langle P(t) \rangle$ with Chebyshev polynomials, estimating their coefficients from the measurements, and then using that information to obtain the first expansion coefficient in the Taylor expansion of $\langle P(t) \rangle$. Then, this information can be used to obtain the coefficients of the Pauli strings comprising the Hamiltonian.

I have not checked the technical correctness of this work. However, the arguments are plausible, and the numerical and analytical results support each other.

For more transparency, the authors should make their code available online.

The discussion of previous work on the topic is brief but seems to be accurate. In short, this work significantly improves upon previous work.

The presentation is not appropriate for any publication. From the main text alone, one can only extract little information on the analytical results that goes beyond the information already contained in the abstract. A lot of deferred definitions are used all over the manuscript, mostly without proper referencing. I think that deferred definitions should only be used with great care, especially when aiming at a high-quality manuscript, as required by Nature Communications. In particular, all quantities used in the main text have to be properly defined in the main text, at least so that experts can understand the definition and with references to where they are defined precisely. The same holds for all other sections. Based on that, also all statements should be self-contained in a similar way. Theorem environments should be self-contained to the extent reasonably possible. Below, I have

compiled a list of minor comments that should help the authors to better understand my criticism of the presentation.

Additionally, a table with short definitions of the relevant quantities and references to the rigorous definitions, e.g. for τ , K , N , A , L , ϵ , δ , r , \mathcal{D} , would be helpful. I also recommend providing a short version of the experimental protocol in the main text, e.g., using an algorithm environment.

Despite this criticism, the technical results seem to all be appropriately discussed somewhere. The only thing missing seems to be an explanation of how the initial states are prepared, say from the $|0\rangle^{\otimes n}$ computational basis state. This should not change the mentioned complexities; however, it should be properly discussed.

I would like the authors to consider using "sampling complexity" instead of "query complexity", which would make the bounds more clear and sound slightly stronger. At least, "query complexity" should be defined because there are different notions (e.g., one can take into account for how long the Hamiltonian is applied or not).

In conclusion, this work presents new and valuable results for Hamiltonian learning. Indeed, the results can be quite useful for experimentalists and theorists alike, but only when presented properly. I am not convinced that the format constraints of Nature Communications allow for an appropriate presentation. I recommend giving the authors the opportunity for a major revision.

Comments

I have compiled this list while reading the manuscript. That is, some of the questions are answered later on but should still be taken into consideration by the authors to make the respective parts of the manuscript more self-contained.

- I think that it is misleading to call $f_{\text{SPAM}}(x)$ SPAM parameters. Usually, SPAM parameters refer to parameters that are independent of the evolution time. However, here $f_{\text{SPAM}}(x)$ is the actual signal that we want to measure.

- The main text seems to be not self-contained, as indicated by the following points.

- What does "up to an error ϵ " mean? More generally, in Sec. II, in what norm are the errors quantified?

- In the beginning, it remains unclear how much control on (ρ_0, O) is needed, i.e., how complex these objects are.
- It needs to be clarified in the main text how the structure parameter D is defined.
- p.3, "This bound is similar in spirit to the shadow norm in shadow tomography": I do not understand this statement. The shadow estimator is unbiased, and the shadow norm is only used to capture the statistical estimation errors, whereas (4) corresponds to systematic ones. I suggest explaining in a different way what the structure parameter is.
- p.3: What is the set $\{X/2, Y/2, Z/2, I/2\}^{\otimes n}$?
- In "After the SPAM parameters have been determined, we then evaluate f_{SPAM} to extract the necessary information about c_1 ", what is the difference between the SPAM parameters and f_{SPAM} evaluated at the relevant points?
- Sec. II.B: What does "structure of the Hamiltonian" mean?
- The main text can and often should contain theorems to concisely outline the main result(s).
- Eq. (5): Sometimes "query complexity" for continuous time evolution takes into account for how long the Hamiltonian is queried. What is the definition here?
- Caption for Fig.2: What is N and τ ?
- Sec. II.D: Can't (7) be simulated efficiently via Jordan-Wigner and free fermions? Why is the TEBD method required?
- Sometimes, the term "shot requirements" is used for the number of samples. It would be good to unify the language.
- What is the reason for choosing $\epsilon=0.021$?
- The numerical work does not seem to be reproducible with the provided information. I recommend writing another appendix section on that.
- Also, the summary in the discussion is not sufficiently self-contained. In the way it is written, it cannot be understood without having read the other text. It should either be dropped entirely or improved.
- Section III: what is k ? Here, also a comparison to the geometric locality, which I think is required by Franca et al., could be made to outline the strength of this work.
- Caption of Fig. 4: "difference between" should probably be "quotient of". The definition of K is said to be contained in Theorem 2, which I cannot follow.

- Definition 1: The type of data access should be part of the definition.
- Before (11): "define the degree of the Hamiltonian D " should be "define the degree D of the Hamiltonian".
- Section IV.A: The statements are general, which is good. However, it should be said at the beginning of the subsection how a general P connects to the main protocol. Otherwise, the reader gets concerned that he or she has to implement general observables.
- The inequality before Theorem 1 also follows from a simple Hoelder inequality.
- Theorem 1: Why is this a theorem and not a lemma? The bound (17) can be wrong if the spectral norm of P is not (sub)normalized. The punctuation around all equations should be checked, also in the appendix.
- Algorithm 1 is referenced before it is stated. Again, in Algorithm 1, Definition 4, Theorem 5 and Lemma 6 are used before they are stated.
- p. 8, Footnote 2: I think that the actual reason for the tildes is that these quantities are estimators, i.e., in particular, random variables.
- It was really annoying when reading the manuscript not to know what ρ and P are for the first 10 pages or so. Often, I would stop reading a paper in such situations.
- \mathbb{V} is not defined.
- Often, the authors say "failure probability" when they mean "maximum failure probability". Again, in Theorem 2, a condition such as $\|P\| \leq 1$ is required. The statement "Then there is some choice of $A \sim \tau$ and $L \sim \log \epsilon^{-1}$ " should be made more precise in Theorem 10.
- What is a "portion of a density matrix"? A reduced state, I guess.
- Lemma 1: "where" should probably be replaced by "such that". There seems to be something wrong with Eq. (26) because ρ_0 appears on both sides.
- Algorithm 2, Line 5: provide a reference to the other algorithm.
- Before Definition 5: improve "see (appendix)".
- Theorem 3: "For the appropriate choice of Chebyshev degree L and evolution time A " could be made more precise in a theorem in the appendix. In the proof, improved "of these sets (see)". Moreover, the computational complexity of finding the sets should be discussed.
- The subscript of, e.g., the sum in (A2) is a somewhat strange notation.
- In a paper, one should only use full equations and avoid a presentation of calculations such

as in the equations in the proof of Lemma 4.

- Before Theorem 5, the $\|b_\ell\|$ are used before they are specified. The presentation should be improved. Similar issues can be found in other places.

- The explanation of how the sampling complexity bound for the median-of-means estimator is used can be slightly extended.

Reviewer #3 (Remarks to the Author):

The authors present a Hamiltonian learning protocol for quantum many-body systems that relies on Chebyshev regression to fit polynomials to time-evolved data. The proposed protocol differs from prior works by generalizing the underlying structure of the Hamiltonian from k -local to sparsely interacting systems. The authors demonstrate their protocol on an 80-qubit transverse field Ising model numerical simulation.

The paper is well written and comprehensive. The main results, however, appear to come from the definition of sparsity, in that the number of coefficients that one needs to determine are reduced even further. As such there is no real parallelization gain from the papers discussed by the authors. The claimed gain from $O(16^k)$ to $O(D^2)$ is not so impressive when one realises that the previous papers described the regime of $D = 4^k$ and that the D parameter is determined by the ansatz of the interactions rather than the protocol itself. Although it is a neat generalisation from local Hamiltonians a little bit of motivation as to why Hamiltonians might be expected to be sparse (as opposed to local) would not go amiss. However it appears there is no “additional” gain in the proposed techniques — it’s just there are fewer parameters to ascertain.

The definition of sparseness, how that can be used to reduce number of parameters and the fitting process described (using Chebyshev regression as opposed to robust polynomial regression) does differ from previous work and is likely to be of independent interest to scientists in the field. The use of graph coloring to complete the proofs relating to the sparseness of the model is particularly nice.

The numerical simulations support the paper; however, they are not particularly experimentally motivated, the 80 qubit system is described using only 159 parameters. It

seems, to us, unlikely that any current 80 qubit system could be accurately modelled by such a Hamiltonian. There is no real discussion as to how this might be robust if there were state preparation and measurement errors, and/or how the models might degrade if the proposed Hamiltonian does not match the actual underlying Hamiltonian.

While the paper is undoubtedly of merit, it is likely to be of most interest to a specialised readership and, therefore, we don't believe it would be a good fit for Nature Communications. It is our recommendation that the authors consider a more specialised journal.

We also note the following:

- on page 2 (top) the claim is for $O(\epsilon^{-1} \text{polylog}(n/\epsilon))$ scaling - should it be ϵ^{-2} ? We can't find anywhere in the text where the ϵ scaling is improved in this way.
- the caption of figure 2 is a little bit terse. Presumably L has to take integer values. N isn't defined in the caption. For the graph on the right is this assuming the optimal values of L and A as set on on the graphs on the right. Is there any intuition as to why it takes 10^{-6} shots to estimate with an error of 10^{-1} Is the error additive or multiplicative?
- The use of SPAM to mean state preparation and measurement settings, rather than state preparation and measurement errors caused us (and we suspect will cause others) a bit of cognitive dissonance. Would it be possible to find a different way to talk about the settings?

Referee 1

I find the work a valuable contribution to the field. Hamiltonian learning is indeed a fundamental and technologically increasingly relevant task. The authors’ approach neatly reduces the amount of samples required for Hamiltonian learning by exploiting the simultaneous measurability of local observables. Using Chebyshev regression instead of finite differences improves robustness and gives analytically well-controlled errors in terms of the polynomial degree and the maximal evolution time. The manuscript is well-written and clear in its presentation. The technical results and proofs in the supplemental material seem sound and correct. One might argue that the authors’ scheme mostly combines multiple known techniques and that the resulting performance guarantees have little surprise value. As such they constitute mostly technical advances that are predominantly of interest for people working on rigorous guarantees for quantum system identification. For this reason, I am not entirely sure if the Nature Communications are the right venue for this work. Having said this, I personally consider the achieved improvement over existing guarantees an important contribution and in principle support a publication in the Nature Communications after a revision. My main point of criticism is that the authors are imprecise and sloppy when contextualizing their contribution (in particular, in the introduction and title). In my opinion, the authors unnecessarily stretch the notion of “practicality” and of a “black-box”. I will explain my criticism below together with further remarks. I would like to encourage the authors to address these points in a revision and present their contribution more carefully.

We appreciate the referee’s feedback and their specific points of criticism regarding the contextualization of our contribution. We have taken referee’s remarks into consideration and made revisions to address these concerns more carefully. To summarize our revisions, we have:

- a) Clarified the distinction between demonstrating the practicality of our scheme and deriving performance guarantees with practical scaling.
- b) Emphasized the improvements in scaling and performance guarantees compared to existing protocols, specifically highlighting the reduction in the scaling of the parameter \mathcal{D} from \mathcal{D}^{21} to \mathcal{D}^4 .
- c) Provided additional explanations and justifications for claiming practical relevance, such as the limitations of previous methods in terms of scaling and unexplored performance on larger systems.
- d) Acknowledged the platform-dependent nature of practicality considerations in experimental implementations and the challenges associated with state preparation, Hamiltonian dynamics, and measurements in experiments. We have also reframed our motivations by highlighting the characterization of near-term quantum computers as a main application for Hamiltonian learning. This perspective helps explain and justify the choices we made in defining “practical” assumptions, which are aligned with the needs and constraints of these experimental systems.

A. “Practicality”

I think one should make a clear distinction between demonstrating that a protocol is practical or deriving performance guarantees for a protocol with ‘practical scaling’. Furthermore, if one does not carefully compare a lower bound for one protocol with an upper bound of another protocol, it is up to debate if one achieved a practically relevant improvement or demonstrated differences in the available proof techniques for the protocols.

The main work we are comparing to is Tang 2021, the point being we improved the scaling in \mathcal{D} from $\mathcal{D}^{21} \rightarrow \mathcal{D}^4$. We agree that in the ideal case, claiming practically relevant improvements would come with something like numerics which can provide evidence for this improvement beyond just asymptotic scaling. However, their paper does not include numerics; we believe it is because the temperatures required are so high that it is actually impractical to do numerical simulations (refer to Eqn. (101) and Eqn. (116) in Tang 2021; setting $\mathcal{D} = 2$ results in a temperature $T \approx 10^8$ and $\mathcal{D} = 3$ results in $T \approx 10^{10}$). These impractically high temperatures are why we claim a practically relevant improvement, as the temperature and time evolution prescriptions in our work are closer to experimental feasibility. To be more concrete, in our work, we show that $A \approx 0.25\tau$ (see Fig. 3) is sufficient, where A is maximum evolution time for unitary dynamics and maximum β for Gibbs states. For unitary dynamics, $\tau = (2\mathcal{D})^{-1}$, and for Gibbs states, $\tau = (2e^2(\mathcal{D}^2 - 1))^{-1}$, see Eqn. (E5). Then we get a maximum $\beta \approx 0.002$ for $\mathcal{D} = 3$. For Chebyshev degree 6, this leads to a minimum $\beta \approx 0.00015$, hence a maximum temperature $T \approx 7000$. This is a 6 orders of magnitude improvement.

When stating their contribution the authors constantly conflate these two notions to argue for the practicality of their scheme. I agree that the authors' scheme seems to be an experimental practical proposal and the scheme comes with 'practical scalings'. However, while the latter point is clearly demonstrated in the manuscript, the former is much less obvious to me.

We slightly weaken the last statement of the abstract to read "Thanks to these improvements, our protocol has practical scaling for large problems: we demonstrate this with a numerical simulation of our protocol on an 80-qubit system."

To give concrete examples where I consider the manuscript not sufficiently precise when comparing to existing results:

- What does it mean to have "scaling results that would allow them to be applied on larger systems"? What exactly limits the application of the techniques of Ref. [13-16] to larger systems? Is their computational complexity problematic or were they simply not demonstrated to succeed?

They were simply not demonstrated to succeed (i.e., no understanding for how well these protocols will perform for system sizes beyond those used for numerics). The text has been updated to reflect this: "Subsequent approaches successfully employed machine learning on small systems. Nonetheless, these methods lacked rigorous performance guarantees or scaling results that would provide confidence in their application to larger systems, as their performance on such systems has not been explored beyond limited numerical studies."

- The authors criticize Refs. [20, 21] for having "constant prefactors that renders them infeasible in practice". Are these prefactors only in the theoretical guarantees and potentially a proof artifact or do they actually determine the practical performance? Is this clearly demonstrated somewhere? When stating their own results in big-O notation the authors also suppress all constant prefactors. So in what sense are they improving? One might even argue that for intermediate system sizes also median-of-mean adds an impractical prefactor that increases the sample complexity by two orders of magnitude.

The constant prefactors are actually prefactors they suppress in their notation (i.e., the dependence on \mathcal{D}). We discuss this in more detail further on in the manuscript, after we have defined what \mathcal{D} is. The scaling with \mathcal{D}^{21} is not just an artifact of the proof, and will show up in practice because it means one must use extremely high temperature Gibbs states in the case of Haah et al. 2022. With regards

to the median-of-means, we have found median-of-means to be unnecessary for our protocol (a simple sample mean suffices) and therefore have removed it.

- When comparing to Ref. [27] in the conclusion it is not clear what k is.

k is the locality of the Hamiltonian in Franca 2022. A generic k -local Hamiltonian can have $\mathcal{D} = 4^k$ (which means every possible k -local Pauli is present in the Hamiltonian). Of course, this is not generally the case in physical Hamiltonians, which is why the case $\mathcal{D} \ll 4^k$ is relevant.

- Regarding the claim of experimental practicality: In my experience, practicality very much depends on the platform, especially when probing an only partly controlled system or an analog simulator. It is a priori not clear if interleaving the time evolution with quenches performing Pauli rotations is prohibitively impractical. At the same time, precisely controlling the transition from state preparation to Hamiltonian dynamics and to a measurement in the end has already been found challenging in experiments. In analogy, in the context QPT, SPAM errors have turned out to severely limit the practicality of many schemes.

We appreciate the referee’s perspective on the practicality considerations in experimental implementations. Indeed, the feasibility and practicality of our proposed techniques can vary depending on the specific platform and level of control. One of the key applications we feel our work is relevant for is the characterization of near-term quantum computers. In this context, we assume the ability to prepare product states and perform time evolution, as it is a reasonable assumption for platforms like Rydberg atom arrays where such operations can be straightforwardly implemented. By considering platforms that allow for state preparation and time evolution, we aim to address the characterization needs of these experimental systems. We mention this platform dependence in the introduction: “We note that the practicality of these assumptions depends on the experimental platform. Indeed, there are other approaches that impose even more stringent assumptions...”

- The numerical demonstration on a 80-qubit transverse field Ising model is a convincing proof-of-concept for the scalability of the authors’ method. But the numerical study is too narrow, does not compare against other methods or systematically studies the effect of experimental imperfections to justify the claim of practicality of the scheme.

We appreciate the referee’s feedback regarding the numerical demonstration in our paper. We agree that while the numerical study provides a convincing proof-of-concept for the scalability of our method, it has certain limitations. Firstly, we acknowledge that our numerical study does not compare our method against other existing methods. We agree that a thorough comparison with alternative approaches would further strengthen the evaluation of our proposed technique. However, due to the features of our method and the lack of comparable techniques in terms of the considered assumptions and scaling guarantees, conducting a direct comparison is challenging. The most directly comparable method is Tang 2021, but their temperature requirements ($T \approx 10^{10}$) render simulations of their protocol infeasible due to numerical imprecision. Secondly, indeed our numerical study does not systematically investigate the effects of experimental imperfections on the practicality of our scheme. We agree that understanding the robustness of our method to realistic experimental conditions is an important aspect to make a strong claim to experimental practicality. We acknowledge this limitation in our manuscript (see Discussion) and emphasize the need for future investigations to address this aspect.

For these reasons, I think the actual practicality in an experiment of the authors’ protocols is not convincingly demonstrated (as also the authors discuss at the end of the conclusion to some extent). At the same time, the authors’ work can nonetheless rightfully claim practical relevance.

We thank the reviewer for their comment. We believe we have made the language in our text more precise regarding the practicality of our protocol, and under what experimental settings we expect our protocol to be useful (i.e., the characterization of near term quantum devices).

B. “Black-box” and “classical interaction with the system”

I find calling the setting a “black-box model” “restricted to classical interaction” misleading. The authors assume accurate control of quantum input (product states) and specific local measurements at the end. In the context of QPT this is commonly referred to as a “prepare-and-measure” setup (say, without entangled input or entangling measurements). The Hamiltonian learning setting additionally assumes control over the evolution time (or inverse temperature). While certainly realistic on many quantum devices today, this setup is still far away from a black-box model with only classical interaction to my understanding. Especially, in the context of Hamiltonian learning it is quite natural to assume settings with considerably more limited control over the input states and the accessible measurements. In fact, this has already motivated an entire series of work by Burgarth and co-authors in the past on inferring Hamiltonians under limited access to the system, for example only measurements on one single local site. The work by Wilde et al. 2022 only requires local measurements but only one initial state, to mention another example.

This is addressed at the end of the introduction. “There are other approaches that impose even more stringent assumptions, such as the restriction of only being able to prepare a single fixed initial state [wilde2022], or the ability to make measurements on only a single site [burgarth2009]. In our work, we do not impose such restricted assumptions, as they do not align with one of the major applications of Hamiltonian learning, namely the characterization of quantum computers. In this context, it remains a natural assumption that we have the ability to prepare arbitrary product states and perform Pauli measurements on arbitrary sites.”

C. Further remarks

To my understanding many ‘natural’ Hamiltonians might have multiple scales. For example, coupling terms might be orders of magnitude suppressed compared to the free dynamics. How does the scheme perform in such scenarios?

We would like to thank the referee for bringing up this interesting point. If we model the Hamiltonian as $H_0 + H_1$, where H_1 is some perturbative Hamiltonian, a naive application of the protocol will require that $\epsilon <$ the scale of H_1 if we want to learn both H_0 and H_1 . Whether we can use less resources to learn H_0 is unclear, and will be an interesting problem for future work!

page 2, first line: I think there is a crucial typo in the stated scaling and it should be $\frac{1}{\epsilon^2}$.

Fixed.

page 1, Even though a common place, saying that QST is exponentially expensive is a bit misleading in the context of Hamiltonian learning. The authors’ guarantee provides a precise estimate in ℓ_∞ -norm (vector max-norm) for the Hamiltonian parameters. Using the same norm, the entries of a quantum state can also be estimated from $\text{polylog}(d)$ state copies using a maximal set of MUBs [Morris & Dakic 2019]. The same holds for spectral norm estimates [Guta et al. 2020]. Only the translation to trace-norm introduces an

exponential scaling in the system size.

It is certainly true that estimating the Hamiltonian parameters using the ℓ_∞ -norm in QST can be done efficiently. For QST applied to Hamiltonian learning, even if we only want the parameters to an error in the ℓ_∞ norm, this requires accuracy of the recovered state in the trace norm (not just the ℓ_∞ norm). This is clarified in the manuscript.

page 3: When the structure parameter \mathcal{D} is first mentioned and used for stating results in Eq. (4) it is not clear what it actually refers to.

A table has been added clarifying all the notation used, along with references to a formal definition.

page 3 typo “initializing the system a product state”.

Fixed.

As far as I know, the first author’s surname of Ref. [27] is Stilck-Franca.

Verified with author that name as written is the preferred way.

page 10, missing reference in proof of Thm. 3.

Fixed.

Referee 2

For any Hamiltonian that is a linear combination of Pauli strings, this work provides an efficient protocol for the estimation of its coefficients via measuring expectation values of time-evolved observables. More specifically, the initial states are certain maximally mixed states in some stabilizer subspaces specifically designed for the task. For the time evolution, the Hamiltonian is applied for certain short time steps. After that, it is sufficient to measure single-qubit Pauli observables. It is proven that the sampling complexity of the estimation protocol scales as (31) with classical postprocessing time (32) where

- \mathcal{D} is a certain degree of the Hamiltonians interaction graph in the Pauli basis,
- $1 - \delta$ is the confidence lower bound for the estimation
- ϵ the maximum ℓ_∞ -norm error of the Hamiltonian coefficients, and
- I cannot find r again while now writing the report.

In particular, these costs are independent of the number of qubits. Moreover, it is shown that the scaling can be improved for commuting Hamiltonians and extended to imaginary time, i.e., to the estimation of thermal states with control in the inverse temperature instead of time. The protocol is also practically applicable, as shown with numerical simulations. Technically, the protocol relies on approximating the expectation values of the time-evolved observables $\langle P(t) \rangle$ with Chebyshev polynomials, estimating their coefficients from the measurements, and then using that information to obtain the first expansion coefficient in the Taylor expansion of $\langle P(t) \rangle$. Then, this information can be used to obtain the coefficients of the Pauli strings comprising the Hamiltonian. I have not checked the technical correctness of this work. However, the arguments are plausible, and the numerical and analytical results support each other.

We thank the reviewer for their comments and clear summary of the manuscript (we note that r is the number of Hamiltonian parameters – a table reviewing the various notations has been added near the beginning of the paper for readability).

For more transparency, the authors should make their code available online.

The code is now published online, see <https://github.com/andigu/hamiltonian-learning>.

The presentation is not appropriate for any publication. From the main text alone, one can only extract little information on the analytical results that goes beyond the information already contained in the abstract. A lot of deferred definitions are used all over the manuscript, mostly without proper referencing. I think that deferred definitions should only be used with great care, especially when aiming at a high-quality manuscript, as required by Nature Communications. In particular, all quantities used in the main text have to be properly defined in the main text, at least so that experts can understand the definition and with references to where they are defined precisely. The same holds for all other sections. Based on that, also all statements should be self-contained in a similar way. Theorem environments should be self-contained to the extent reasonably possible. Below, I have compiled a list of minor comments that should help the authors to better understand my criticism of the presentation.

We have aimed to make the main text as self-contained as possible. References to future definitions, theorems, etc. have been eliminated to the furthest extent possible by either removing the references where they do not add much to the text, or moving definitions and theorems into the main text where appropriate. For instance, we have moved our ‘main result’ (Theorem 1) into the main text, along with two crucial definitions, namely Definition 1, which formalizes the Hamiltonian learning problem, and Definition 2 which describes the interaction graph and the meaning of \mathcal{D} .

Additionally, a table with short definitions of the relevant quantities and references to the rigorous definitions, e.g. for $\tau, K, N, A, L, \epsilon, \delta, r, \mathcal{D}$, would be helpful. I also recommend providing a short version of the experimental protocol in the main text, e.g., using an algorithm environment.

Added (Table 1 and Algorithm 1).

Despite this criticism, the technical results seem to all be appropriately discussed somewhere. The only thing missing seems to be an explanation of how the initial states are prepared, say from the $|0\rangle^{\otimes n}$ computational basis state. This should not change the mentioned complexities; however, it should be properly discussed.

We clarify that “More specifically, the reduced state for each site will be either the maximally mixed state $I/2$ or an eigenstate of $X, Y,$ or Z ; the full state ρ_0 is a tensor product of these single qubit states. These states are easily prepared from $|0\rangle^{\otimes n}$ by applying a constant number of single qubit gates.” For instance, we can apply a Hadamard gate on sites where we require an eigenstate of X , or a Hadamard followed by a phase gate for sites where we require eigenstates of Y . The maximally mixed state is easily prepared by simply randomly applying an X gate with 50% probability.

I would like the authors to consider using “sampling complexity” instead of “query complexity”, which would make the bounds more clear and sound slightly stronger. At least, “query complexity” should be defined because there are different notions (e.g., one can take into account for how long the Hamiltonian is applied or not).

All occurrences of “query complexity” have been replaced with “sample complexity”.

In conclusion, this work presents new and valuable results for Hamiltonian learning. Indeed, the results can be quite useful for experimentalists and theorists alike, but only when presented properly. I am not convinced that the format constraints of Nature Communications allow for an appropriate presentation. I recommend giving the authors the opportunity for a major revision.

We thank the reviewer for their comments on the presentation of our manuscript. We believe the revised version is significantly more readable and fits the format constraints of Nature Communications much better, in large part thanks to the reviewer’s suggestions.

I think that it is misleading to call $f_{\text{SPAM}}(x)$ SPAM parameters. Usually, SPAM parameters refer to parameters that are independent of the evolution time. However, here $f_{\text{SPAM}}(x)$ is the actual signal that we want to measure.

The subscript SPAM is to indicate that we have a collection of functions f_1, f_2, \dots that are indexed by the various possible SPAM settings. We have used the notation \mathcal{S} to denote one particular instance of SPAM setting, and written the function as $f_{\mathcal{S}}$ instead. We hope this is clearer.

The main text seems to be not self-contained, as indicated by the following points. What does “up to an error ϵ ” mean? More generally, in Sec. II, in what norm are the errors quantified?

This is now addressed in the notation table, and reiterated in the main text.

In the beginning, it remains unclear how much control on (ρ_0, O) is needed, i.e., how complex these objects are.

We added clarification “However, we do not insist on arbitrary control over ρ_0 and O ; we only consider the case where ρ_0 is fully separable and O being a local Pauli operator.”

It needs to be clarified in the main text how the structure parameter \mathcal{D} is defined.

It is now mentioned in the table of notation and a formal definition is linked.

p.3, “This bound is similar in spirit to the shadow norm in shadow tomography”: I do not understand this statement. The shadow estimator is unbiased, and the shadow norm is only used to capture the statistical

estimation errors, whereas (4) corresponds to systematic ones. I suggest explaining in a different way what the structure parameter is.

We removed reference to shadow tomography, and added the clarification “... and also depends on the assumptions we make about the Hamiltonian (i.e., the structure parameter \mathcal{D} , whether the Hamiltonian is commuting)”

p.3: What is the set $\{X/2, Y/2, Z/2, I/2\}^{\otimes n}$?

It has been explicitly laid out what the state ρ_0 will be: “and the initial state ρ_0 will be a fully separable state. More specifically, the reduced state for each site will be either the maximally mixed state $I/2$ or an eigenstate of X , Y , or Z ; the full state ρ_0 is a tensor product of these single qubit states.”

In “After the SPAM parameters have been determined, we then evaluate f_{SPAM} to extract the necessary information about c_1 ”, what is the difference between the SPAM parameters and f_{SPAM} evaluated at the relevant points?

f_{SPAM} is $\text{Tr}(P e^{-iHt} \rho_0 e^{iHt})$. The subscript SPAM sets P and ρ_0 , and evaluating it at different times just gives the value of the observable at different times. The subscript SPAM was changed to \mathcal{S} , perhaps this resolves any confusion about what the SPAM parameters correspond to.

Sec. II.B: What does “structure of the Hamiltonian” mean? The main text can and often should contain theorems to concisely outline the main result(s).

It means that these constants depend only on \mathcal{D} . This is now clarified.

Eq. (5): Sometimes “query complexity” for continuous time evolution takes into account for how long the Hamiltonian is queried. What is the definition here?

Query complexity is replaced with sample complexity, which should clarify that this does not take into account evolution time. If we include evolution time, the query complexity is actually improved by a factor $1/\mathcal{D}$ compared to the query complexity, because the evolution time goes as $1/\mathcal{D}$.

Caption for Fig.2: What is N and τ ?

This is now clarified in table of notation.

Sec. II.D: Can’t (7) be simulated efficiently via Jordan-Wigner and free fermions? Why is the TEBD method required?

This specific Hamiltonian can indeed be simulated with JW and free fermions. However, the code (published on Github) was written to allow a general (sparsely interacting) Hamiltonian to be simulated.

Sometimes, the term “shot requirements” is used for the number of samples. It would be good to unify the language.

We removed “shot requirements” in favor “sample complexity”.

What is the reason for choosing $\epsilon = 0.021$?

A rounder number $\epsilon = 0.01$ was chosen in the new version. Results may look slightly different because of the removal of the median-of-means estimator.

The numerical work does not seem to be reproducible with the provided information. I recommend writing another appendix section on that.

The code has been published (<https://github.com/andigu/hamiltonian-learning>). All figures in the paper should be reproducible with the published code.

Also, the summary in the discussion is not sufficiently self-contained. In the way it is written, it cannot be understood without having read the other text. It should either be dropped entirely or improved.

If this refers to the summary of our contribution (i.e., the first paragraph of the discussion), we have expanded the text in places where it previously may have been unclear. It has been done with the aim to

make the summary more self-contained. On the other hand, if it refers to our comparison with other works, we have given brief explanations of how our results improve upon other works' results.

Section III: what is k ? Here, also a comparison to the geometric locality, which I think is required by Franca et al., could be made to outline the strength of this work.

A brief explanation of k has been added, and a discussion of the geometric locality assumption has been included.

Caption of Fig. 4: “difference between” should probably be “quotient of”. The definition of K is said to be contained in Theorem 2, which I cannot follow. K is no longer a necessary parameter as a result of the removal of the median-of-means estimator.

Definition 1: The type of data access should be part of the definition.

Fixed.

Before (11): “define the degree of the Hamiltonian \mathcal{D} ” should be “define the degree \mathcal{D} of the Hamiltonian”.

Fixed.

Section IV.A: The statements are general, which is good. However, it should be said at the beginning of the subsection how a general P connects to the main protocol. Otherwise, the reader gets concerned that he or she has to implement general observables.

We added clarification early on in the text: “. . . , for P being a local Pauli operator, . . .”

The inequality before Theorem 1 also follows from a simple Hoelder inequality.

This is indeed a simpler way of seeing this fact. The von Neumann inequality is replaced with the Hoelder inequality now.

Theorem 1: Why is this a theorem and not a lemma? The bound (17) can be wrong if the spectral norm of P is not (sub)normalized. The punctuation around all equations should be checked, also in the appendix.

Theorem 1 has been changed to a lemma, and the condition $\|P\| \leq 1$ has been added. Punctuation around equations has been updated.

Algorithm 1 is referenced before it is stated. Again, in Algorithm 1, Definition 4, Theorem 5 and Lemma 6 are used before they are stated.

Algorithm 1 has been simplified and the references in question were either removed (as they became unnecessary) or rearranged so that they are stated beforehand.

p. 8, Footnote 2: I think that the actual reason for the tildes is that these quantities are estimators, i.e., in particular, random variables.

This footnote has been removed. We felt it was not useful in clarifying the idea, at least in the main text.

It was really annoying when reading the manuscript not to know what ρ and P are for the first 10 pages or so. Often, I would stop reading a paper in such situations.

It is now clarified on pg. 4 that “. . . the observables O_i can be chosen to be single-qubit Paulis and the initial state ρ_0 will be a fully separable mixed state. More specifically, the reduced state for each site will be either the maximally mixed state $I/2$ or an eigenstate of X , Y , or Z ; the full state ρ_0 is a tensor product of these single qubit states.”

\mathbb{V} is not defined.

We replaced \mathbb{V} with var (indicating variance) - this is probably more widely recognized.

Often, the authors say “failure probability” when they mean “maximum failure probability”. Again, in Theorem 2, a condition such as $\|P\| \leq 1$ is required. The statement “Then there is some choice of $A \sim \tau$ and $L \sim \log \epsilon^{-1}$ ” should be made more precise in Theorem 10.

Failure probability has everywhere been replaced with “maximum failure probability” or “failure proba-

bility at most δ ". The condition on $\|P\|$ has been added, and the precise choices of A and L are given in the statement of Theorem 10.

What is a "portion of a density matrix"? A reduced state, I guess.

Yes, the language has been changed to clarify this.

Lemma 1: "where" should probably be replaced by "such that". There seems to be something wrong with Eq. (26) because ρ_0 appears on both sides.

The $\rho_0^{(z)}$ that appears on the right is merely to indicate that the density matrix on the 'Z' sites can be completely arbitrary.

Algorithm 2, Line 5: provide a reference to the other algorithm.

Added.

Before Definition 5: improve "see (appendix)".

Fixed.

Theorem 3: "For the appropriate choice of Chebyshev degree L and evolution time A " could be made more precise in a theorem in the appendix. In the proof, improved "of these sets (see)". Moreover, the computational complexity of finding the sets should be discussed.

The choice of A and L (or at least their asymptotic scaling) is indicated in Theorem 10. In practice, A and L are found numerically by optimizing the error bounds (see caption of Figure 2). The complexity of finding the sets is included now in the main text.

The subscript of, e.g., the sum in (A2) is a somewhat strange notation.

This has been fixed for Eqns. (A2) - (A5). For sums such as in Eqn. (B5), this notation is the most compact way of expressing the constraint over the summed variables.

In a paper, one should only use full equations and avoid a presentation of calculations such as in the equations in the proof of Lemma 4.

The calculations that do not contain useful techniques and are just 'plug and chug' (especially those in the section about Chebyshev polynomials) have been removed or significantly abbreviated. Other calculations which we feel to contain some insight have been retained.

Before Theorem 5, the b_ℓ are used before they are specified. The presentation should be improved. Similar issues can be found in other places.

We clarify that "...where the coefficients $b_\ell \in \mathbb{R}$ are parameters that we fit to the data." Similar issues elsewhere have been addressed.

The explanation of how the sampling complexity bound for the median-of-means estimator is used can be slightly extended.

Upon review, it turns out that a median-of-means estimator is overkill; a simple sample mean estimator suffices.

Referee 3

The paper is well written and comprehensive. The main results, however, appear to come from the definition of sparsity, in that the number of coefficients that one needs to determine are reduced even further. As such there is no real parallelization gain from the papers discussed by the authors. The claimed gain from $O(16^k)$ to $O(\mathcal{D}^2)$ is not so impressive when one realises that the previous papers described the regime of $\mathcal{D} = 4^k$ and that the \mathcal{D} parameter is determined by the ansatz of the interactions rather than the protocol itself. Although it is a neat generalisation from local Hamiltonians a little bit of motivation as to why Hamiltonians might be expected to be sparse (as opposed to local) would not go amiss. However it appears there is no “additional” gain in the proposed techniques — it’s just there are fewer parameters to ascertain.

The gain is not just in the fact that we assume fewer parameters, so less work is required. It is that we are able to provide tighter bounds on things such as the n th order time derivative of observables; bounds like Lieb-Robinson bounds can only give bounds that go like $\exp(k)$, regardless of whether every possible k -body Pauli is present, or if only a couple are. By being more precise about the Hamiltonian structure and using more of our prior knowledge, we are able to improve this bound. Indeed, in the worst case, the two bounds are the same, but the point is that experimentally, we can often rule out certain terms (e.g., from symmetry considerations), so experimentally relevant Hamiltonians are often quite sparse (i.e., $\mathcal{D} \ll 4^k$). This is expanded on further in the discussion.

The definition of sparseness, how that can be used to reduce number of parameters and the fitting process described (using Chebyshev regression as opposed to robust polynomial regression) does differ from previous work and is likely to be of independent interest to scientists in the field. The use of graph coloring to complete the proofs relating to the sparseness of the model is particularly nice. The numerical simulations support the paper; however, they are not particularly experimentally motivated, the 80 qubit system is described using only 159 parameters. It seems, to us, unlikely that any current 80 qubit system could be accurately modelled by such a Hamiltonian. There is no real discussion as to how this might be robust if there were state preparation and measurement errors, and/or how the models might degrade if the proposed Hamiltonian does not match the actual underlying Hamiltonian.

A discussion on the experimental relevance of transverse field Ising models is included in the paper. Replicated here, “We choose the TFIM for its broad range of applications, including its relevance for quantum computing platforms such as Rydberg atom arrays.” In short, the Rydberg Hamiltonian is essentially a transverse field Ising model with tunable couplings (see <https://iopscience.iop.org/article/10.1088/2058-9565/aa9c59>).

While the paper is undoubtedly of merit, it is likely to be of most interest to a specialised readership and, therefore, we don’t believe it would be a good fit for Nature Communications. It is our recommendation that the authors consider a more specialised journal.

We respectfully disagree with reviewer. Hamiltonian learning, the process of discerning an unknown Hamiltonian through measurements, has gained paramount significance. It serves as a vital prerequisite for validating quantum simulators and multi-qubit devices in general. Given the ongoing transformation of noisy quantum computers into scientific instruments, the field of Hamiltonian learning holds relevance across a wide spectrum of audiences. Papers solving different problems in the field of Hamiltonian learning have been published in high profile journals before. Here are few examples:

- “Learning a Local Hamiltonian from Local Measurements” by E. Bairey et al., Physical Review Letters

- “Quantum Variational Learning of the Entanglement Hamiltonian” by C. Kokail et al., Physical Review Letters
- “Suppressing Qubit Dephasing Using Real-Time Hamiltonian Estimation” by M. Shulman et al., Nature Communications
- “Hamiltonian Learning and Certification Using Quantum Resources” by N. Wiebe et al., Physical Review Letters
- “Learning Many-Body Hamiltonians with Heisenberg-Limited Scaling” by H. Huang et al., Physical Review Letters
- “Experimental Quantum Hamiltonian Learning” by J. Wang et al., Nature Physics

Vis-a-vis the improvements of our protocol, see the discussion to the first point. In short, the well-behaved scaling with \mathcal{D} is important because

- \mathcal{D} is often much smaller than 4^k . For future works that make use of this insight, our work contains useful theoretical perspectives. For instance, we believe the tools we use to bound things in terms of \mathcal{D} (e.g., graph coloring) will be of interest for a wide audience.
- Other works that give bounds in terms of \mathcal{D} (e.g., Haah, Kothari, and Tang 2021) have bounds that scale like \mathcal{D}^{21} – this is a completely impractical bound. In contrast, our scaling with \mathcal{D}^4 is much more manageable, and offers the possibility of implementation in practice. We support this with our numerical simulations.

There is no real discussion as to how this might be robust if there were state preparation and measurement errors, and/or how the models might degrade if the proposed Hamiltonian does not match the actual underlying Hamiltonian.

A discussion on these two problems is included in the final paragraph of the “Discussion” section as directions for future investigation.

on page 2 (top) the claim is for $O(\epsilon^{-1}\text{polylog}(n/\epsilon))$ scaling - should it be ϵ^{-2} ? We can't find anywhere in the text where the ϵ scaling is improved in this way.

Fixed.

the caption of figure 2 is a little bit terse. Presumably L has to take integer values. N isn't defined in the caption. For the graph on the right is this assuming the optimal values of L and A as set on on the graphs on the right. Is there any intuition as to why it takes 10^{-6} shots to estimate with an error of 10^{-1} Is the error additive or multiplicative?

The definitions of N and L are included in a table of notations on pg. 3. The fact that L (in practice) has to take on integer values has been added. The intuition is that the rough breakdown is, a factor $\sim 10^2$ comes from shot noise, and the remaining factor of 10^4 has to do with the fact that we need to compensate for the bias in our derivative estimator. That is, unlike the case of something like shadow tomography, we do not have access to an unbiased estimator.

The use of SPAM to mean state preparation and measurement settings, rather than state preparation and measurement errors caused us (and we suspect will cause others) a bit of cognitive dissonance. Would it be possible to find a different way to talk about the settings?

Other readers also noted confusion at the subscript SPAM in f_{SPAM} . We have replaced this with \mathcal{S} to denote the fact that \mathcal{S} encodes a particular choice of state prep plus the measurement.

REVIEWERS' COMMENTS

Reviewer #1 (Remarks to the Author):

I find most of my concerns that were detailed in my previous report sufficiently addressed in the revision of the main text. In addition, the presentation of the manuscript has been significantly improved. My criticism of using the term "black-box" in this context without any justification still applies to the title. In my opinion, the title is misleading and has not much to do with the scientific content of the authors' work. I highly recommend to change it. Under a proper title, I think the work is overall suitable for a publication in the Nature Communications in principle. However, my reservations remain that the work is a borderline case regarding the significance required for such a high-impact journal. It is mainly a technical improvement of one previous result on scaling of recovery guarantees and slightly broader applicable than competing works. Most importantly, the work itself does not make a convincing case for its practical relevance. The robustness of the authors' scheme under practical conditions is not demonstrated at all in the manuscript. The latter actually requires more extensive numerical studies, analyzing more settings and including at least toy models of realistic errors that are unavoidable in an experiment (e.g. the effect of SPAM errors and also errors in the time evolution). Arguably, in the context of quantum characterization, it is common practice to include considerably more detailed numerical studies also in works that mostly focus on proving guarantees. In my opinion this effort is justified, especially when aiming at relevance of the results for a broader audience.

Reviewer #2 (Remarks to the Author):

In my first report, I criticized the presentation of the manuscript. In the revised version, the presentation has been improved a lot. I think that it is now in a suitable form for Nat. Commun. and I recommend its publication.

Also, the code has been made publically available.

Reviewer #3 (Remarks to the Author):

I thank the authors for their considered reply. I believe that the changes made in response to my comments and the more extensive comments from the other reviewers have greatly improved the manuscript. The question then remains as to whether Nature communications is the correct place to publish. There has never been any doubt that Hamiltonian learning is an important topic. The question is whether the advances made in this manuscript are of sufficient interest and innovation to warrant publication in this journal. There are two main claims, improved scaling and improved recovery guarantees. I will deal with these separately:

****Improved scaling****. I read the authors' discussion of this point carefully, but stand by my opinion that the main gains here are primarily a recasting of the problem. The improved scaling comes about through the reduction of parameter count (the sparsely interacting Hamiltonian). Although I stand to be corrected I believe other protocols, with no or minimal changes, would have identical scaling with the same sparsely interacting Hamiltonian assumptions. The parallelisation is a nice application of a technique relating to utilising entire stabiliser sets, but this is now a technique that is being used in many bodies of work. I understand the authors' point that such sparse Hamiltonians may be realistic (e.g. in Rydberg atom arrays), although how general such sparse Hamiltonians might be is a moot point. The demonstration of recovery of 159 parameters in an 80 qubit one-dimensional Ising model may be a good demonstration of the protocol but it is not groundbreaking, other papers have demonstrated their protocol on 100-qubit chains with over 1,000 parameters. I remain slightly puzzled as to why the demonstration is on a one-dimension chain rather than a two dimensional Ising model as in the Rydberg atom array paper the authors rely on. Overall I still applaud the innovations of the authors relating the analysis of scaling and utilisation of graphs, as set out in my initial review, but remain unconvinced that on this point Nature communications is the correct venue for the manuscript.

****Improved recovery guarantees****. This was not a point I focussed on in my initial review, being more concerned with the scaling claims made by the authors, with the guarantees being more in the nature of a technical, rather than practical, improvement. I have read the view of referee 1 with interest. I find myself in agreement with the views they express.

There does indeed appear to be an improvement over existing guarantees and as the authors mention in their reply the methods of extracting the parameters may well be more practical in extracting parameters with lower error bounds than previous methods. I remain concerned as to the impact of state preparation and measurement errors on the recovery, but I am happy to agree that such concerns form future work. On this front I will support publication in Nature communications.

Two final minor points for the authors to consider.

- 1) Out of context the claim on page 2 of "samples to recover every parameter of an n-qubit Hamiltonian up to an error epsilon", might be a bit bold. I presume here they mean a sparsely interacting Hamiltonian as later defined.
- 2) The use of k for k-local and as the derivative coefficient count (e.g. equation 9) is potentially confusing. I assume this is why it doesn't appear in the Glossary.

Referee 1

I find most of my concerns that were detailed in my previous report sufficiently addressed in the revision of the main text. In addition, the presentation of the manuscript has been significantly improved. My criticism of using the term "black-box" in this context without any justification still applies to the title. In my opinion, the title is misleading and has not much to do with the scientific content of the authors' work. I highly recommend to change it. Under a proper title, I think the work is overall suitable for a publication in the Nature Communications in principle. However, my reservations remain that the work is a border-line case regarding the significance required for such a high-impact journal. It is mainly a technical improvement of one previous result on scaling of recovery guarantees and slightly broader applicable than competing works. Most importantly, the work itself does not make a convincing case for its practical relevance. The robustness of the authors' scheme under practical conditions is not demonstrated at all in the manuscript. The latter actually requires more extensive numerical studies, analyzing more settings and including at least toy models of realistic errors that are unavoidable in an experiment (e.g. the effect of SPAM errors and also errors in the time evolution). Arguably, in the context of quantum characterization, it is common practice to include considerably more detailed numerical studies also in works that mostly focus on proving guarantees. In my opinion this effort is justified, especially when aiming at relevance of the results for a broader audience.

We would like to thank the Referee for the time dedicated to reviewing our manuscript and providing insightful feedback. We concur with Referee's observation regarding the use of the term "black-box" in the title. Recognizing its potential misalignment with the core content of our research, we have changed our title to "Practical Hamiltonian Learning with Unitary Dynamics and Gibbs States", and hope the Referee will agree that this more accurately reflects the key contributions of our work. Regarding the concerns about the practical relevance and robustness of our scheme, we understand their significance, especially in the context of Hamiltonian learning. While our primary focus has been on proving guarantees, we acknowledge the value of detailed numerical studies in demonstrating the practical applicability of our results. For the scope of this submission, we have chosen to emphasize the foundational and theoretical aspects of our approach. However, Referee's suggestions for more extensive numerical studies are genuinely appreciated, and while they might not be incorporated in this particular manuscript, they will be considered in our future work. In conclusion, we hope that the title change, along with the revisions made, will address Referee's concerns and position our work as suitable for publication in Nature Communications. Once again, we would like to thank the Referee for constructive feedback.

Referee 2

In my first report, I criticized the presentation of the manuscript. In the revised version, the presentation has been improved a lot. I think that it is now in a suitable form for Nat. Commun. and I recommend its publication.

We would like to thank the Referee for acknowledging the improvements made in the revised manuscript. We appreciate Referee's recommendation for publication in Nature Communications. Referee's feedback has been instrumental in refining our work.

Referee 3

I thank the authors for their considered reply. I believe that the changes made in response to my comments and the more extensive comments from the other reviewers have greatly improved the manuscript. The question then remains as to whether Nature communications is the correct place to publish. There has never

been any doubt that Hamiltonian learning is an important topic. The question is whether the advances made in this manuscript are of sufficient interest and innovation to warrant publication in this journal. There are two main claims, improved scaling and improved recovery guarantees. I will deal with these separately: ****Improved scaling****. I read the authors' discussion of this point carefully, but stand by my opinion that the main gains here are primarily a recasting of the problem. The improved scaling comes about through the reduction of parameter count (the sparsely interacting Hamiltonian). Although I stand to be corrected I believe other protocols, with no or minimal changes, would have identical scaling with the same sparsely interacting Hamiltonian assumptions. The parallelisation is a nice application of a technique relating to utilising entire stabiliser sets, but this is now a technique that is being used in many bodies of work. I understand the authors' point that such sparse Hamiltonians may be realistic (e.g. in Rydberg atom arrays), although how general such sparse Hamiltonians might be is a moot point. The demonstration of recovery of 159 parameters in an 80 qubit one-dimensional Ising model may be a good demonstration of the protocol but it is not groundbreaking, other papers have demonstrated their protocol on 100-qubit chains with over 1,000 parameters. I remain slightly puzzled as to why the demonstration is on a one-dimension chain rather than a two dimensional Ising model as in the Rydberg atom array paper the authors rely on. Overall I still applaud the innovations of the authors relating the analysis of scaling and utilisation of graphs, as set out in my initial review, but remain unconvinced that on this point Nature communications is the correct venue for the manuscript.

We would like to thank the Referee for continued engagement and detailed feedback on our manuscript. We genuinely appreciate the depth of Referee's review and the concerns that Referee raised. Regarding the improved scaling, while we acknowledge the perspective the Referee has shared, we emphasize that our gains are not purely derived from a recasting of the problem. Our approach integrates multiple techniques. When combined, they offer a distinct advantage in Hamiltonian learning. The sparsely interacting Hamiltonian is indeed a significant aspect of our work, but it is one of several components that contribute to the improved scaling. On the topic of the one-dimensional Ising model, we chose this model not merely for simplicity but because it remains experimentally realistic and relevant (for instance, most state-of-the-art neutral atom arrays have ~ 100 qubits). While other papers have explored larger qubit chains or more parameters, our focus was on demonstrating the efficiency and applicability of our protocol in a setting that aligns with real-world quantum systems.

Our intention is to highlight the unique contributions of our work within the broader context of Hamiltonian learning, and we believe that our manuscript, with its combination of theoretical insights and practical demonstrations, offers a valuable perspective to the community. We hope our clarifications address Referee's concerns.

****Improved recovery guarantees****. This was not a point I focussed on in my initial review, being more concerned with the scaling claims made by the authors, with the guarantees being more in the nature of a technical, rather than practical, improvement. I have read the view of referee 1 with interest. I find myself in agreement with the views they express. There does indeed appear to be an improvement over existing guarantees and as the authors mention in their reply the methods of extracting the parameters may well be more practical in extracting parameters with lower error bounds than previous methods. I remain concerned as to the impact of state preparation and measurement errors on the recovery, but I am happy to agree that such concerns form future work. On this front I will support publication in Nature communications.

We appreciate Referee's thoughtful consideration of the improved recovery guarantees presented in our manuscript. We are happy to hear that our efforts in this direction resonate with both Referee's perspective

and that of Referee 1. Indeed, we believe the methods we have introduced offer a more practical approach to extracting parameters with lower error bounds, which is a significant step forward in the field of Hamiltonian learning. Referee’s concerns regarding the impact of state preparation and measurement errors are valid and critical for the practical implementation of our protocol. Choosing to emphasize the foundational and theoretical aspects of our approach in this work, we view these as essential areas for future exploration and plan to address these challenges in subsequent work.

Two final minor points for the authors to consider.

1) Out of context the claim on page 2 of ”samples to recover every parameter of an n -qubit Hamiltonian up to an error ϵ ”, might be a bit bold. I presume here they mean a sparsely interacting Hamiltonian as later defined. 2) The use of k for k -local and as the derivative coefficient count (e.g. equation 9) is potentially confusing. I assume this is why it doesn’t appear in the Glossary.

We would like to thank the Referee for highlighting these two points. First, we have clarified the context on page 2 to ensure that the claim specifically refers to a sparsely interacting Hamiltonian, as defined later in the manuscript. Second, the potential confusion arising from the use of k for both k -local and the derivative coefficient count was recognized and addressed by replacing k for the derivative coefficient with m .